# Recent Development of Nanomaterials-Based Cytosensors for the Detection of Circulating Tumor Cells

**DOI:** 10.3390/bios11080281

**Published:** 2021-08-18

**Authors:** Zhi-Fang Sun, Yong Chang, Ning Xia

**Affiliations:** College of Chemistry and Chemical Engineering, Anyang Normal University, Anyang 455000, China; allensune@gmail.com (Z.-F.S.); 7180610011@stu.jiangnan.edu.cn (Y.C.)

**Keywords:** circulating tumor cells, cancer biomarkers, electrochemical biosensors, optical biosensors, nanomaterials

## Abstract

The accurate analysis of circulating tumor cells (CTCs) holds great promise in early diagnosis and prognosis of cancers. However, the extremely low abundance of CTCs in peripheral blood samples limits the practical utility of the traditional methods for CTCs detection. Thus, novel and powerful strategies have been proposed for sensitive detection of CTCs. In particular, nanomaterials with exceptional physical and chemical properties have been used to fabricate cytosensors for amplifying the signal and enhancing the sensitivity. In this review, we summarize the recent development of nanomaterials-based optical and electrochemical analytical techniques for CTCs detection, including fluorescence, colorimetry, surface-enhanced Raman scattering, chemiluminescence, electrochemistry, electrochemiluminescence, photoelectrochemistry and so on.

## 1. Introduction

Tumor metastasis results in about 90% of cancer mortality. Since Ashworth first proposed the concept of rare circulating tumor cells (CTCs) in 1869, it has been growingly accepted with increasing evidences during the past ten years [1]. CTCs originate from the primary tumor cells and enter into the peripheral blood. They are the precursors of metastasis and have been found in many cancer types such as breast, prostate, lung and colorectal cancer [2,3,4]. As one form of liquid biopsy, CTCs are considered as the important and promising biomarkers for cancers, which potentially represent global features of cancer characteristics [5,6]. Sensitive analysis and specific identification of CTCs in peripheral blood samples are beneficial to early diagnosis, prognosis and treatment efficacy evaluation of cancers. This has prompted intensive efforts into the development of efficient methods for the capture and detection of CTCs. However, the abundance of CTCs in the blood is extremely low (a few to hundreds per mL) alongside with a large number of hematologic cells (10^9^ cell per mL) [7,8]. Therefore, efficient capture and sensitive detection of CTCs from blood still remain a tremendous challenge.

Due to the complexity of real samples and the low abundance of CTCs, a large panel of techniques have been employed to enrich CTCs from the blood [9,10,11]. Physical enrichment based on their characteristic biophysical properties, such as size, deformability and density, is one family of attractive isolation methods. The sizes of tumor cells in the serum are always larger than those of normal red blood cells and white blood cells. Such a difference provides the possibility to isolate these cells via high-precision filtration [12]. Although size-based isolation methods are label-free, simple and non-invasive, the probable cross in size between CTCs and white blood cells might decrease the purity of enriched samples (normally < 10%). In addition, the difference in deformability and density can also be used to isolate cancer-derived CTCs [13]. The negative surface charge of cancer cells can be utilized as an effective biophysical marker for the isolation and detection of CTCs because of the excretion of lactic anions in glycolysis pathway [14]. Apart from different biophysical properties, protein biomarkers are always over-expressed on the surface of CTCs, which can be used as the targets for the isolation and analysis of CTCs from blood cells through the specific interaction between the membrane proteins and molecular probes, including antibodies, peptides and aptamers [15]. For example, epithelial cell adhesion molecule (EpCAM) is highly expressed on the surface of various epithelial carcinomas [16]. The human epidermal growth factor receptor 2 (HER2) was reported to be useful for the detection of breast cancer CTCs [17]. Prostate specific membrane antigen (PSMA) and prostate-specific antigen (PSA) are two important indicators for the diagnosis of prostate cancer [18,19]. Thus, biorecognition elements towards certain cell phenotype-dependent biomarker are chosen for the detection of cancer cells. Among them, antibodies as the most popular probes are usually employed to develop immunoaffinity-based methods by antibodies-modified magnetic beads (MBs) and nanostructured microfluidic chips [20,21,22,23,24,25,26,27,28]. The MBs-based and microfluidic techniques can be easily integrated with various optical and electrochemical analysis methods, promoting the development of novel biosensors [29,30,31,32]. However, the instability and high cost of antibodies limit their further applications. As the surrogates of antibodies, peptides and aptamers have been widely screened and used to isolate and detect CTCs due to their small size, excellent stability and ease of synthesis and functionalization [33,34,35,36,37,38,39,40]. Furthermore, high level of glycoproteins on the surface of cancer cells can be utilized as the biomarkers for CTCs, which can be recognized by lectin or small molecules with boronic acid groups [41,42,43]. Folate receptor (FR) over-expressed on the surface of cancer cells can specifically bind to folate with high affinity [44]. However, epithelial-to-mesenchymal transitions (EMT) during cancer metastasis may alter the EpCAM expression of CTCs, thereby decreasing the detection sensitivity. Isolation techniques based on single biomarker may result in the escape of CTCs. Thus, utilization of multiple probes toward different biomarkers on the same cell surface can provide a better strategy for enrichment and analysis of CTCs.

To meet the rapid progress in the demand of early diagnosis of cancers, a number of optical and electrochemical sensing techniques have been developed for sensitive and accurate detection of CTCs [4,45]. During the last two decades, a variety of advanced nanomaterials and nanostructures with exceptional physical and chemical properties have been prepared to fabricate cytosensors with hybrid bio/nanostructures for amplifying the signal and enhancing the sensitivity [6,46,47,48,49]. For example, nanomaterials with high surface area-to-volume ratios and excellent electrical conductivity, such as carbon nanotubes (CNTs) and graphene oxide (GO), are used to modify the electrode, improving the capture of CTCs and promoting the electron transfer. Nanomaterials with unique enzyme-like catalytic capabilities are employed to catalyze redox reactions in optical and electrochemical assays. Due to the interesting localized surface plasmon resonance phenomena (LSPR), noble metal nanoparticles (NPs) have been used as optical signal transducers, fluorescence modulation and Raman signal enhancement. Thus, the combination of these highly versatile nanomaterials with various sensing techniques can dramatically improve the isolation efficiency and detection sensitivity.

Recently, a few review articles have been reported on the enrichment and detection methodologies of CTCs [50,51,52,53,54]. For instance, Khademhossieni’s group and Zhou’s group reviewed the progress in electrochemical detection of breast cancer cells and other CTCs [55,56]. Farshchi et al. reviewed recent progress and challenges in microfluidic biosensing of CTCs [57]. Koo et al. summarized the applications of magnetic nanomaterials in electrochemical detection of diverse circulating cancer biomarkers [58]. Furthermore, aptamer-based cytosensing approaches for the capture and detection of CTCs have been covered in several review papers [38,50,52,53]. However, to the best of our knowledge, there are few reports focused on nanomaterials-based cytosensors for CTCs detection including both immunosensors and aptasensors [59,60]. In this review, we aim to give a comprehensive review about various nanoplatforms and highlight the different roles of nanomaterials in cytosensors. For discussion purposes, developments in CTCs detection are classified according to the sensing techniques, which covered mainly optical and electrochemical transducers. The optical cytosensors include fluorescence, colorimetry, surface-enhanced Raman scattering and chemiluminescence. Electrochemical methods are mainly represented by electrochemistry, electrochemiluminescence and photoelectrochemistry. Their analytical performances including the detection limit and linear range are compared in Table 1 and Table 2.

## 2. Nanomaterials-Based CTCs Biosensors

Nanomaterials possess diverse and outstanding characteristics that are different from those of their bulk counterparts. The inclusion of nanomaterials into various biosensors can amplify the signal and improve the sensing performances. For example, nanomaterials with high area surface and excellent conductivity have been introduced into the interface chemistry to accelerate the electron transfer and improve the biomolecule compatibility and affinity.

### 2.1. Fluorescence Cytosensors

Fluorescent biosensors enable highly sensitive and selective detection of many target analytes. Generally, fluorescent tags are required to conjugate with the detection elements for signal transduction. For the application of nanotechnology into the detection of cancer biomarkers, several types of inorganic or organic nanomaterials with excellent photoluminescent properties have been employed to enhance the sensitivity and versatility of fluorescence biosensors [61].

#### 2.1.1. Dye-Doped Nanomaterials

Traditional dyes have several drawbacks, such as photobleaching, weak fluorescence intensity and ease of self-aggregation. To overcome these disadvantages, nanomaterials are utilized to load dyes for enhancing the fluorescence properties and amplifying the signal [62,63]. For example, dye-doped silica nanoparticles (dye-SiNPs) with improved photostability and stable fluorescence signal have been used for the detection of CTCs [64,65,66,67]. Jo et al. synthesized dual aptamer-functionalized dye-SNPs (dual-SiNPs) for simultaneous detection of two types of breast cancer cells [68], in which biotin-modified aptamers were bound to avidin-conjugated SiNPs (Figure 1A). In the presence of MUC1(+) cells (MCF-7) and HER2(+) cells (SK-BR-3), dual-SiNPs were bound to the cells captured by MBs and then detected. Metal-organic frameworks (MOFs) modified with 1,2-dioleoyl-sn-glycero-3-phosphate (DOPA) lipid bilayer (DOPA-LB) were utilized to carry near-infrared dyes for sensitive detection of CTCs in vivo [69].

Due to the improved stability and extended blood circulation time, dye-labeled MOF@DOPA-LB can sensitively detect the early smaller tumor lesion (1–2 mm). For simultaneous detection of two cancer cells, Ho et al. proposed a fluorescence method based on the retro-self-quenching dye-doped 1,3-phenylenediamine resin (DAR) nanoparticles (DAR NPs) (Figure 1B) [70]. The NPs doped with different dyes were modified with two different aptamers (sgc8c and TD05), respectively. The fluorescence of DAR NPs was weak because of the self-quenching. However, when the aptamer-modified NPs bound with and entered into cells, the fluorophores were released from the NPs, leading to the fluorescence recovery.

#### 2.1.2. Luminescent Nanomaterials

To overcome the shortcomings of dyes, various luminescent nanomaterials with excellent photoluminescence properties and biocompatibility have been designed. For example, semiconductor quantum dots (QDs) have been widely utilized in fluorescent biosensors for the detection of CTCs because of their unique properties, such as narrow emission bandwidth, high photostability and excellent fluorescence quantum yield [71,72,73,74]. Hsieh et al. presented a QDs-based fluorescent immunosensor for the detection of low concentration of human Jurkat cells (T-help cells) based on immunomagnetic separation [75]. As shown in Figure 2A, the cells were recognized by biotinylated anti-CD3, which allowed for the modification with streptavidin-modified QDs. Then, anti-CD4-MBs were added to separate the sandwich-like complexes performed between anti-CD3, cells and QDs for fluorescent analysis. To amplify the fluorescent signal during the recognition event, Hua et al. employed silica NPs to load nucleolin aptamer AS1411-modified QDs (Figure 2B) [76]. In this work, MCF-7 breast cancer cells were sensitively detected by fluorescence and square-wave voltametry after magnetic accumulation and fluorescent labeling. Anti-EpCAM antibody-conjugated QDs were used to quantify CTCs with anti-IgG-modified MBs for isolation [77]. Moreover, QDs-embedded poly(styrene/acrylamide) copolymer nanospheres and QDs/Fe_3_O_4_ NPs-encapsulated SiO_2_ NPs were used as the multifunctional nanoprobes for simultaneous capture and authentication of CTCs [78,79]. The one-step detection process not only simplified the detection procedure and shortened the detection time, but also preserved cell viability. DNA probes with high programmability have proven to be a perfect tool for the self-assembly of different nanomaterials. A DNA-templated MNPs-QD-aptamer copolymers (MQAPs) were developed and used for rapid capture and detection of CTCs [80]. As shown in Figure 2C, a polymeric DNA template was first prepared by hybridization chain reaction (HCR). The overhangs in H1 were utilized to conjugate DNA-modified QDs and MNPs with the aid of additional linkers. The overhang in H2 was a DNA aptamer (sgc8c) toward protein tyrosine kinase 7 (PTK7) over-expressed on human leukemia CCRF-CEM (CEM) cells. The resulted MQAPs showed enhanced magnetic response, which promoted the binding selectivity for target cells and increased the fluorescence intensity. Finally, rare CTCs in blood samples were facilely isolated and analyzed within 20 min.

Compared with QDs, gold nanoclusters (AuNCs) exhibit favorable biocompatibility, simple synthetic procedure and stable photoluminescence. Tao et al. proposed a dual-ligand-functionalized, AuNCs-based pattern sensing strategy for the assays of CTCs [81]. As displayed in Figure 2D, seven types of AuNCs were prepared as both effective cell recognition elements and signal transducers. AuNCs with different surface properties interacted with different cell populations with various surface protein and molecules, generating distinct fluorescence responses.

Carbon dots, including carbon quantum dots and graphene quantum dots (GQDs), have drawn numerous attention due to their tunable photoluminescent properties and high photostability. Cui et al. developed a magnetic fluorescent biosensor based on nitrogen and sulfur-doped graphene quantum dots (N,S/GQDs) and molybdenum disulfide (MoS_2_) nanosheets [83]. In this work, GQDs were bound to the aptamer-functionalized magnetic NPs (aptamer@Fe_3_O_4_@GQD) for one-step bioimaging and enrichment, in which MoS_2_ nanosheets were employed to quench the fluorescence. The interaction between the aptamer@Fe_3_O_4_@GQD nanocomposites and CTCs facilitated the release of MoS_2_ nanosheets from the nanocomposites, leading to the restoration of fluorescence signal. Cobalt oxyhydroxide nanosheets have also been used to quench the fluorescence of folic acid (FA)-functionalized carbon dots and label the MBs-enriched cells [84]. In this work, ascorbic acid (AA) was added to decompose the nanosheets, resulting in the fluorescence recovery and achieving the sensitive detection of CTCs.

The self-fluorescence and scattering from the complex sample are the main interference for fluorescence methods. Luminescent nanomaterials with near-infrared (NIR) excitation or emission are widely popular in biosensing and bioimagining. Ding et al. reported a NIR fluorescent signal amplification strategy using Ag_2_S nanodots for magnetic capture and detection of CTCs [82]. As shown in Figure 2E, Ag_2_S nanodot-based probes were produced by DNA-labeled nanodots through HCR. The multi-aptamer in nano-assembly increased the binding ability of nanodots toward cells. Then, anti-EpCAM-modified magnetic nanospheres (MNs) were added to isolate the cells. In their another work, Ag_2_S nanodots with tetra-DNA nanostructures generated in-situ through a “one-pot” method were utilized to determine CTCs with the use of the hybrid cell membrane-coated magnetic nanobioprobes [85]. Multivalent aptamer functionalization and hybrid cell membrane coating greatly enhanced the capture efficiency toward CTCs due to the synergistic effect. Lanthanide-based upconversion nanoparticles (UCNPs) can be excited by multiple low-wavelength NIR photons and emit a high-energy photon at a shorter wavelength. They are the promising alternatives to NIR dyes and QDs because of their excellent photostability and free autofluorescence-background [86]. Fang et al. employed aptamer-conjugated UCNPs to analyze CTCs [87], in which aptamer and biotin-conjugated UCNPs were used to recognize tumor cells and then the UCNPs-cells were captured by avidin-modified MNPs for imaging and analysis.

#### 2.1.3. Nanoquencher-Based Fluorescent Cytosensors

Graphene oxide (GO) and its derivates can quench the fluorescence of dyes and nanomaterials through Forster resonance energy transfer (FRET). A porous GO membrane with a pore size of 20–30 nm was prepared and labeled with three different aptamers for the capture and identification of multiple types of cancer cells (Figure 3A) [88]. In this work, the fluorescence of dye-labeled aptamers was completely quenched by GO. When the cell targets bound to the aptamers, the configuration DNA changed, which resulted in the increase of the distance between dye and GO and the follow-up recovery of fluorescence signal. A reduced graphene oxide (rGO) and aptamer-modified microarray was also constructed to capture and detect cancer cells in a “turn-on” detection format [89]. Moreover, GO can bind with single-stranded DNA via hydrophobic and π-π stacking interactions. Plenty of label-free GO/DNA-based methods have been reported for bioassays. Cao et al. constructed a GO-based label-free cytosensing microfluidic chip for visual and high-through detection of CCRF-CEM cells [90]. However, the non-specific displacement caused by other molecules and incomplete interaction between aptamers and cells reduced the sensitivity of this method. For this view, they further developed a GO-based cytosensor through cyclic enzymatic signal amplification [91]. As shown in Figure 3B, hairpin aptamers were firstly bound to cells and initiated nicking endonuclease-assisted signal amplification. Then, the remanent intact dye-labeled DNA was adsorbed and its fluorescence was quenched by GO. Thus, an increasingly amplified fluorescent signal was observed with the increasing number of target cells. This “turn-on” method had a high sensitivity and a low detection limit down to 25 cells/mL.

### 2.2. Colorimetric Cytosensors

As an apparatus-free detection approach, colorimetric assay has been developed to determine different targets. Combined with various powerful DNA-based signal amplification strategies, colorimetric cytosensors have been constructed for CTCs detection by using peroxidase-mimicking G-quadruplex DNAzymes [92,93,94,95]. Nanomaterials play different roles in colorimetric assays as the colorimetric substrates, the carriers of dyes or natural enzymes and the nanozymes with powerful catalytic ability.

Nanomaterials with a large surface area can load plenty of chromogenic compounds and controllably release them for signal amplification. A pH-sensitive allochroic dye-based nanobioplatform was proposed for simultaneous capture and colorimetric detection of two heterogeneous CTCs (MCF-7 and HeLa) [96]. As shown in Figure 4, polyacrylic acid (PAA)-modified molybdenum disulfide nanoflakes (MoS_2_NFs) were utilized to load two kinds of aptamers (SYL3C and C−9S) and pH indicators (thymolphthalein (TP) and curcumin (CUR)). Meanwhile, multivalent aptamers-AuNPs were used as the supporting substrates to increase the capture efficiency because of their multiple binding sites and high affinity toward CTCs. After the formation of sandwich-like nanostructures, the entrapped dye molecules were transformed into color soluble ion species and then released into the solution under the alkaline condition (pH 12.5). The color of TP changed from colorless to dark blue with the appearance of an absorption peak at 590 nm and that of CUR changed from colorless to bright yellow accompanied with a new absorption peak at 470 nm. This method allowed for the detection of CTCs down to 5 cells/mL with naked eye.

Due to the unique LSPR properties, Au and Ag nanomaterials possess high extinction coefficients and show distance, morphology and size-dependent color changes. Therefore, they have been widely introduced into colorimetric cytosensors instead of traditional dyes. Aggregation-based colorimetric sensing approach is the most popular mode. In 2008, Medley reported a simple AuNPs-based colorimetric method for the detection of cancerous cells (Figure 5A) [97]. Aptamer-conjugated AuNPs could specifically bind to the target cells and the assembly/aggregation of AuNPs around cells resulted in a red shift of LSPR maximum absorption and an obvious change of solution color.

Since Fe_3_O_4_ NPs were found to exhibit intrinsic peroxidase-like activity in 2007 for the first time, numerous nanomaterials have been exploited with enzyme-like properties under the near physiological conditions, which are coined as “nanozymes” [98,99]. Kip et al. developed a simple colorimetric method for CTCs detection based on a cell internalizable nanozyme (Figure 5B) [100]. Fe_3_O_4_ NPs were immobilized within the porous interiors of silica microspheres and further modified with hyaluronic acid (HA) (HA@Fe_3_O_4_@SiO_2_ microspheres). Microspheres could be uptaken by human cervical cancer (HeLa) cells and primary brain tumor cells (T98 G glioblastoma) via pinocytosis, resulting in the decrease of nanozyme concentration in solution and the weakness of the corresponding absorption peak.

Generally, the nanozyme-catalytic reactions happen on the nanostructure surface. Thus, one can tune the catalytic activity of nanozymes by adjusting the surface chemistry. For example, Zhu et al. developed an immunomagnetical colorimetric method for CTCs detection by using DNA to enhance the peroxidase activity of single-walled carbon nanotubes (SWCNTs) [101]. As shown in Figure 5C, biotinylated EpCAM aptamer (P1) was modified on the surface of “core-shell” immunomagnetic nanoparticles (Fe_3_O_4_–SiO_2_-Gel) and then hybridized with three messenger DNA (mDNA) strands. After the specific interaction between CTCs and aptamers, mDNA strands were released and wrapped onto the surface of SWCNTs, resulting in the enhancement of the peroxidase activity of SWCNTs. The captured cells released by the degradation of gelatin through MMP-9 could be subsequently recultivated. In addition, NPs with other novel catalytic properties also attracted numerous attentions. Sun et al. found that the nanocomposites of Pd NPs and covalent organic framework (COF) could promote the cleavage of *N*-butyl-4-NHAlloc-1,8-naphthalimide (NNPH) into *N*-butyl-4-amido-1,8-naphthalimide (NPH), alongside with significant changes of solution color and fluorescence intensity [102].

Up to date, some artificial peroxidase-like small molecules have been discovered and applied in colorimetric assays, including fluorescein and 10-methyl-2-amino-acridone [103,104]. Huang et al. for the first time found that ethylene diaminetetraacetic acid (EDTA) exhibits an interesting photocatalytic property under light irradiation, which can catalyze the decomposition of hydrogen peroxide (H_2_O_2_) into oxidative OH radical [105]. Based on the fascinating property, they constructed a plasmonic colorimetric biosensor for visual detection of MCF-7 cells by using gold nanorods (GNRs) as the indicators (Figure 5D). The anisotropic GNRs suspensions displayed various colors with the change of the aspect ratio of GNRs. After the immunoreaction, EDTA conjugated with the secondary antibody catalyzed the production of hydroxyl radical, which regulated the aspect ratio of GNRs and resulted in the change of solution color.

**Figure 5 biosensors-11-00281-f005:**
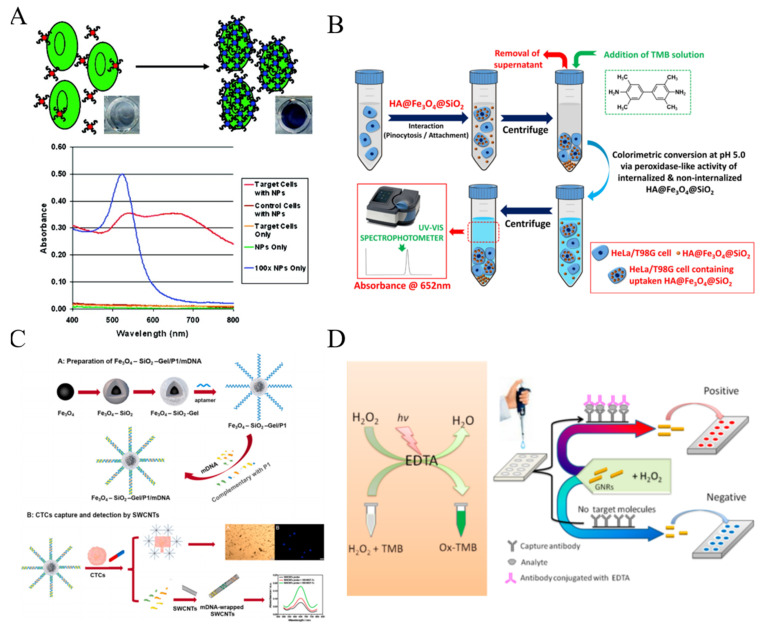
(**A**) Schematic representation of the ACGNP based colorimetric assay [97]. Copyright 2008 American Chemical Society. (**B**) Schematic representation of colorimetric protocol developed for determination of HeLa or T98 G cell concentration based on the variation of peroxidase-like activity of cell internalizable HA@Fe_3_O_4_@SiO_2_ microspheres [100]. Copyright 2020 Elsevier B.V. (**C**) Schematic representation of Fe_3_O_4_–SiO_2_-Gel/P1/mDNA, CTCs capture and subsequent colorimetric detection of the captured CTCs by SWCNTs colorimetric probe [101]. Copyright 2021 Elsevier B.V. (**D**) Schematic representation of visual read-out for determination of CTCs by combining EDTA-based sensor with immunomagnetic separation of CTCs from blood [105]. Copyright 2017 American Chemical Society.

### 2.3. SERS Cytosensors

Raman spectroscopy is a vibrational optical spectroscopy based on the inelastic scattering of light by a target molecule. The intrinsically low intensity of Raman scattering can be enhanced when the analyte is placed near a rough metal surface. This approach, called surface enhanced Raman spectroscopy (SERS), holds huge potential for trace analysis and surface science due to its advantages of single molecule sensitivity, resistance to photobleaching and narrow spectral bandwidth. Moreover, near-infrared excitation allows the application of SERS cytosensors in biological samples, such as whole blood.

Plasmonic Au and Ag-based nanomaterials are the most widely used materials in SERS cytosensors [106,107,108,109,110,111,112]. In 2008, Sha et al. developed the SERS AuNPs-based biotags for the detection of breast cancer cells [113]. As shown in Figure 6A, anti-EpCAM-modified MBs were used for the capture of CTCs in whole human blood and anti-HER2 antibody-modified SERS biotags were employed for the signal readout. The detection limit was calculated to be about 50 cells/mL, with 99.7% confidence in whole blood. However, the enrichment step for CTCs capture in whole blood made the detection procedure complicated. To simplify the detection procedure, the direct detection of target CTCs in blood have attracted wide interest [114]. For example, Wu et al. proposed a new SERS nanoplatform for the direct detection of CTCs in rabbit blood [115]. As shown in Figure 6B, AuNPs were successively modified with 4-mercaptobenzoic acid (4-MBA), reductive bovine serum albumin (rBSA) and FA. The thin layer of rBSA had a lower weakening effect on the SERS signal than thick poly(ethylene glycol) (PEG) protection layer. This method had a linear detection range of 5−500 cells/mL and a detection limit of 5 cells/mL. The shape of gold nanomaterials also plays a vital role in SERS. Generally, asymmetric shape of nanomaterials can provide a more intensive electromagnetic (EM) enhancement [116]. Core-shell plasmonic nanorods with tunable and linker-free nanogaps, triangular silver nanoprisms and Au nanoflowers were also utilized to selectively determine CTCs [117,118,119,120]. Nima et al. proposed a multicolor SERS method for CTCs identification [121]. As illustrated in Figure 6C, Ag-Au NRs with narrow SERS spectral lines were functionalized with four Raman-active reporters and four different antibodies, respectively. Based on the antibody rainbow cocktail and significant signal enhancement from these Raman probes, single breast cancer cell has been selectively detected in unprocessed whole human blood. Moreover, the nanocomposites can be used to construct SERS substrate for the capture and detection of cancer cells, such as Ag nanowires-rGO [122].

Hot spots at the junctions between neighbor NPs can induce dramatic SERS enhancement due to the coupling of LSPR. For this view, the controllable AuNPs-based assembly is a promising approach for sensitive SERS assays. Zhang et al. developed a novel SERS probe by assembling AuNPs in triangular pyramid DNA (TP-AuNPs) with strong electromagnetic hot spots for enhancing Raman scattering [123]. The synthesis process was illustrated in Figure 6D. This novel SERS probe had intense plasmonic hot spots and definite binding site. The method showed a linear detection range of 3–500 cell/mL without enrichment process.

### 2.4. Chemiluminescence Cytosensors

Chemiluminescence is the luminescence generated from the chemical reaction without the use of an excitation light. Due to the advantages of simple instrumentation, no scattering light interference and high sensitivity, chemiluminescence has been widely utilized in the fields of food safety and clinical diagnosis [124,125]. Nanomaterials with different chemical composition have been widely introduced into systems for enhancing chemiluminescence intensity as nanocatalysts or nanocarriers.

Nanocatalysts can increase the luminescence yield, such as AuNPs and carbon-based nanomaterials, due to the increased surface area and high electron density [126]. Cao et al. proposed an enzyme-free chemiluminescence strategy for CTCs detection based on Au@luminol NPs, hybridization chain reaction (HCR) and magnetic isolation [127]. As illustrated in Figure 7, AS1411 nucleolin aptamer-modified MNPs were utilized to capture and isolate MCF-7 cells from whole blood samples. Then, the resulted Au@luminol-HCR assemblies (ALHA) were used to specifically recognize the cells for the generation of intensive chemiluminescence signal with K_2_S_2_O_8_ as the co-reactant. Based on the amplification effect, the detection sensitivity was down to 3 cells/mL. Large amounts of metal ions could be released from metal NPs by oxidative metal dissolution to act as catalysts. Bi et al. developed a synergistically enhanced chemiluminescence system based on the dual-amplification of MNPs [128]. After the DNA amplification and the formation of multiplex nanoprobe (CuS/DNA/Au/DNA/MNP), numerous Cu^2+^ and Fe^3+^ ions were released to catalyze the luminol-H_2_O_2_ chemiluminescence reaction. This method showed a detection limit of 56 cells/mL. Similarly, Ru^3+^ released from RuNPs was also employed to construct luminol–H_2_O_2_–Ru^3+^ chemiluminescence system for cancerous cell detection [129].

**Figure 7 biosensors-11-00281-f007:**
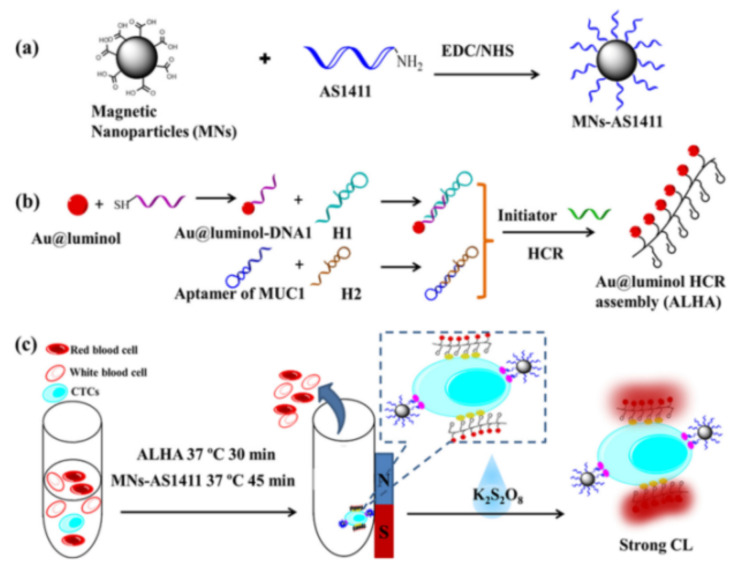
Schematic of experimental process: (**a**) Preparation of AS1411 modified Fe_3_O_4_ nanoparticles (MNs-AS1411), (**b**) Preparation of Au@luminol-HCR assembly (ALHA) and (**c**) Specific recognition, isolation and CL detection of CTCs by MNs-AS1411/ALHA system [127]. Copyright 2020 Elsevier B.V.

**Table 1 biosensors-11-00281-t001:** Analytical performances of various nanomaterials-based optical techniques for CTCs detection.

Detection Method	Type of Nanomaterials	Detection Limit (Cells/mL)	Linear Range (Cells/mL)	Ref.
Fluorescence	SiNPs	10	Not reported	[68]
	DAR NPs	44 for CCRFCEM; 79 for Ramos	1.5 × 10^4^–7.5 × 10^4^	[70]
	QDs	Not reported	40–1 × 10^5^ cells	[72]
		390	1 × 10^2^–5 × 10^5^	[77]
	QDs-loaded SiNPs	201	250–1 × 10^4^	[76]
	DNA-assisted Ag_2_S nanoassembly	Not reported	10–500	[82]
	CDs	5	10–1 × 10^4^	[84]
	rGO	22	1 × 10^2^–2 × 10^4^	[89]
	GO	25	25–2.5 × 10^4^	[90]
		25	50–1 × 10^5^	[91]
Colorimetry	MoS_2_ NFs	2 for HeLa; 4 for MCF-7	5–1 × 10^4^	[96]
		90	2 × 10^2^–4 × 10^4^	[97]
	Fe_3_O_4_ NPs-SiNPs	Not reported	0.25–4 × 10^6^	[100]
	MNPs	13	50–5 × 10^4^	[99]
	PtAu NPs	10	10–1 × 10^6^	[95]
	Pd NPs/CMC-COF	100	1 × 10^2^–1 × 10^6^	[102]
	SWCNTs	3	10–500	[101]
Surface enhanced Raman scattering	AuNPs	<10	Not reported	[113]
		Not reported	5–500	[114]
		5	5–500	[115]
	AgNPR	5	5–100	[118]
	AuNFs	5	5–200	[119]
	Au-AgAu core-shell structure	Not reported	5 × 10^2^–3 × 10^4^	[117]
	Au@Ag NRs	1	1–100	[120]
	AuNPs	Not reported	3–500	[123]
Chemiluminescence	AuNPs	163 cells	0–2 × 10^3^	[125]
		3	10–1 × 10^5^	[127]
	CuS/DNA/Au/DNA/MNP	56	80–1 × 10^3^	[128]
	RuNPs	62	1 × 10^2^–1 × 10^3^	[129]

Abbreviates: SiNPs, silica nanoparticles; DAR NPs, 1,3-phenylenediamine resin nanoparticles; QDs, quantum dots; CDs, carbon dots; rGO, reduced graphene oxide; GO, graphene oxide; MoS_2_ NFs, molybdenum disulfide nanoflakes; MNPs, magnetic nanoparticles; CMC, carboxymethyl cellulose; COF, covalent organic framework; SWCNTs, single-walled carbon nanotubes; AuNPs, gold nanoparticles; AgNPR, triangular silver nanoprisms; AuNFs, gold nanoflowers; Au@Ag NRs, gold-silver nanorods.

### 2.5. Electrochemical Cytosensors

Electrochemical biosensors have attracted considerable attention in recent years because of their high sensitivity, simple operation and rapid responsive time [130]. The bio-interface and signal reporter are two critical factors for electrochemical biosensors. According to the detection format, electrochemical biosensors can be classified into two types: direct detection and sandwich-like detection.

#### 2.5.1. Direct Detection

As a noninvasive detection mode, electrochemical impedance spectroscopy (EIS) exhibits a high potential for cell-related application in a narrow range of applied potentials because of its advantages of low cost, label-free procedure and easy operation [131,132]. Cells captured by the sensor electrode can prevent the electron transfer of electroactive substance in solution, which increases the charge transfer impedance. Normally, nanomaterials with large surface area and high electricity property can be utilized to modify the electrode as the support materials, thus improving the conductivity and sensitivity.

Carbon nanotubes (CNTs) have attracted considerable attention in the development of electrochemical biosensors because of their high electrical conductivity and excellent absorption ability [133]. For example, β-cyclodextrin-modified MWCNTs were used to identify cancer cells by modification with a tetrathiafulvalene derivative [134]. The negatively charged SWNTs induced the assembly of positively charged ferrocene-functionalized poly(ethylene imine) on the surface of electrode via the layer-by-layer technique, which allowed for the detection of FR-mediated HeLa cells [135]. Xu et al. developed a cytosensor with 3D-MWCNTs array based on vicinal-dithiol-containing proteins (VDPs) [136]. As shown in Figure 8A, MWCNTs were immobilized on the indium tin oxide (ITO) electrode as the nanostructured surface. 2-*p*-Aminophenyl-1,3,2-dithiarsenolane (VTA2) modified on the MWCNTs selectively reacted with vicinal dithiol in vicinal-dithiol-containing protein (VDP), which was over-expressed in cancer cells. Polydopamine (PDA) was used to encapsulate CNTs and immobilize FA for cell identification [137]. HA conjugated on the surface of positively charged poly(diallyldimethylammonium chloride)-modified CNTs by the electrostatic interaction could specifically recognize CD44 by ligand-protein interaction [138].

Graphene, a two-dimensional (2D) single sheet composed of sp^2^-hybridized carbon atoms, has been applied in the biosensing [139]. The poly-L-lysine/GO-modified electrode has been utilized for the adhesion and detection of leukemia K562 cancer cells through the electrostatic interaction [140]. Li et al. reported an electrochemical biosensor for the detection of cancer cells based on FA and octadecylamine-modified graphene aerogel microspheres [141]. As displayed in Figure 8B, GO sheets in the emulsion were self-assembled into graphene oxide gel microspheres on the water/toluene interfaces. After freeze-drying and reduction by H_2_, the formed FA-GAM-OA exhibited a sphere-like structure with plenty of open-pores and FA groups. Then, the microspheres were deposited on the electrode for CTCs detection with FA as the capture probe.

In addition to the good biocompatibility, noble metal NPs can accelerate the electron transfer between NPs-loaded materials and electrode. Thus, noble metal NPs have been intensively used to construct electrochemical biosensors [142,143,144]. Seenivasan et al. employed anti-MC1R-Ab-labeled AuNPs to detect CTCs by differential pulse voltammetry (DPV) [145]. Au nanowire arrays were prepared by electrochemical deposition with anodic aluminum oxide as template for the capture and release of CCRF-CEM cells [146]. A galactosylated gold-nanoisland biointerface was fabricated for the capture and detection of HepG-2 cells though the multivalent galactosyl interaction [147]. Under light irradiation, the excited LSPR around AuNPs surface can generate a strong electromagnetic field and a high concentration of electron−hole pairs, which could enhance the electrochemical performances [148]. Wang et al. developed a direct plasmon-enhanced electrochemical cytosensor for the label-free detection of CTCs in blood [149]. As shown in Figure 8C, Au nanostars (AuNSs) were coated on the surface of glassy carbon (GC) electrode and then modified with aptamer sgc8c, which could specifically recognize the transmembrane receptor protein tyrosine kinase 7 on CCRF-CEM cell membrane. Upon light illustration, the current was significantly enhanced with AA as the electroactive probe. After incubation with CTCs, the current signal decreased significantly due to the blocking effect. This method achieved a low detection limit of 5 cells/mL. Sun et al. proposed a competitive electrochemical aptasensor for HepG2 tumor cells based on hybrid nanoprobes of Pd-Pt nanocages labeled with complementary DNA, hemin/G-quadruplex DNAzyme and HRP [150]. Moreover, AuNPs-based nanocomposites with other nanomaterials, such as monodispersed carbon nanospheres and Prussian blue analogues were prepared and used for the detection of CTCs [151,152].

**Figure 8 biosensors-11-00281-f008:**
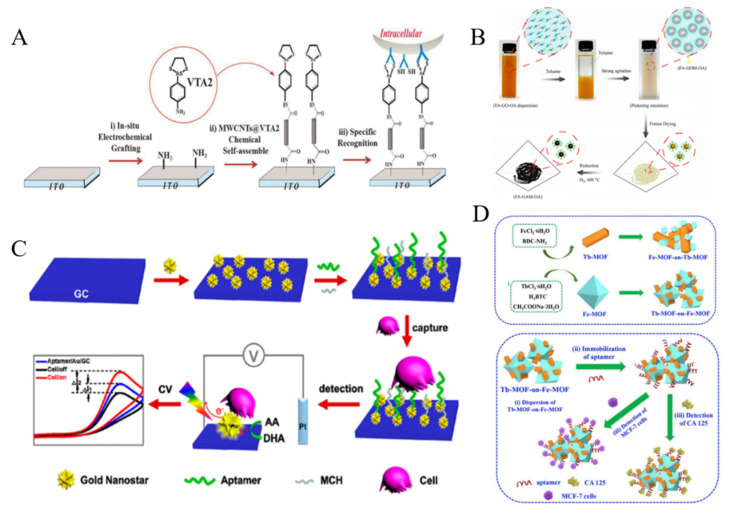
(**A**) Illustration of f assembling processes of 3D-MWCNTs@VTA2 cytosensor for detection of VDPs overexpressed tumor cell [136]. Copyright 2015 Elsevier B.V. (**B**) Illustration of the procedure for synthesis of FA-GAM-OA [141]. Copyright 2018 Elsevier B.V. (**C**) Schematic representation of the strategy for ultrasensitive and label-free detection of CTCs by the DPEE mechanism [149]. Copyright 2019 American Chemical Society. (**D**) Schematic representation of the preparation of Fe-MOF-on-Tb-MOF and Tb-MOF-on-Fe-MOF nanostructures and the fabrication procedure of aptasensor based on two kinds of TbFe-MOFs [153]. Copyright 2019 Elsevier B.V.

Metal-organic frameworks (MOFs) consisted of metal ions and organic ligands are a class of well-defined porous crystalline nanomaterials. Due to their high surface area, tunable pore sizes and abundant compositions, MOFs have been broadly used in biosensing and biocatalysts. Du et al. proposed an impedance cytosensor based on FA-functionalized MOFs (UiO-66) [154]. The detection limit was calculated to be 90 cells/mL. Versatile properties can be integrated into heterostructured MOFs via synergistic interaction. Wang et al. synthesized bimetallic TbFe-MOFs by MOF-on-MOF strategy and employed them to detect MCF-7 cells (Figure 8D) [153]. They found that Tb-MOF-on-Fe-MOF displayed a better performance than Fe-MOF-on-Tb-MOF for CTCs detection. In this work, aptamers were adsorbed on the surface of MOFs for the capture of cells due to their excellent adsorption ability and functional groups.

Covalent organic frameworks (COFs), an emerging class of porous crystalline polymers, have aroused wide interests of various fields. Yan et al. used 2D porphyrin-based COF to immobilize EGFR-targeting aptamer for MCF-7 cell detection [155]. Mesoporous silica NPs, MoS_2_ nanosheets, Prussian blue analogue and butterfly-like TiO_2_ nanomaterials have been functionalized with recognition elements for cell detection [149]. In recent years, bio-nanomaterials assembled by peptides are increasingly attractive due to their biocompatibility and facile synthesis. Peptide nanoparticles and nanotubes were prepared by self-assembly of diphenylalanine-based peptide and used to develop electrochemical platforms for cell detection [156,157].

#### 2.5.2. Sandwich-Like Detection

As one of the most widely used detection formats, sandwich-like cytosensor shows high sensitivity and selectivity. Generally, enzymes, electroactive molecules and functional nanomaterials have been broadly utilized to label recognition elements for signal amplification [158,159,160,161]. For example, Yu et al. proposed a simple alkaline phosphatase (ALP)-based electrochemical aptamer cytosensor for the detection of cancer cells [162]. Ferrocene-labeled concanavalin A was employed for detecting K562 cells based on the lectin–carbohydrate interaction [163]. However, the instability of enzymes and weak electrochemical signal of small molecules limited their applications. To overcome these problems, various functional nanomaterials are developed to enhance the performances of electrochemical cytosensors. According to the roles in signal amplification, these nanomaterials can be classified into three groups: carriers for signal molecules, nanoelectrocatalysts and electroactive probes.

It is a general approach to achieve efficient signal amplification by loading a great number of signal probes or enzymes onto nanomaterials [164,165,166]. Ding et al. employed Con A and HRP-labeled AuNPs to detect CTCs based on the glycan-lectin interaction [167]. As shown in Figure 9A, single-walled carbon nanohorns (SWNHs) were used to modify the electrode for improving electrical connectivity, which were further functionalized with arginine-glycine-aspartic acid-serine tetrapeptide to recognize K562 cell. Con A on the AuNPs interacted with mannose oligosaccharide on the cell surface, and HRP on the AuNPs catalyzed the reaction to generate a strong electrochemical signal. Moreover, tumor necrosis factor-related apoptosis-inducing ligands and aptamers have been loaded on HRP-modified AuNPs for the detection of CTCs [168,169]. Nanohybrids of AuNPs with other nanomaterials were utilized to carry HRP and HRP-mimicking DNAzyme (hemin/G-quadruplex) for CTCs detection, such as ZnO nanorods and MOFs [170,171,172]. Ou et al. developed a sandwich-type cytosensor for capture, detection and release of breast cancer cells based on HRP and PtNPs-loaded MOF [173]. As displayed in Figure 9B, dual aptamers (AS1411 and MUC1) were immobilized on the electrode with tetrahedral DNA nanostructures (TDN). MOF PCN-224 was decorated with PtNPs and HRP molecules. Meanwhile, two capture probes consisted of the sequence of AS1411 or MUC1 aptamer and G-quadruplex were modified on the MOF. In the presence of hemin and K^+^ ions, the catalytic HRP-mimicking G-quadruplex/hemin (GQH) DNAzymes were generated. After the formation of sandwich-like structure, multifunctional hybrid nanoprobes containing PCN-224-Pt, HRP and GQH DNAzyme catalyzed the generation of benzoquinone (BQ) from the oxidation of HQ with H_2_O_2_. In addition to enzymes, electroactive small molecules can also be modified on nanomaterials to detect CTCs [174,175,176]. For instance, Zhang et al. reported a lectin-based biosensor for the assays of A549 cells based on thionine (Th)-labeled AuNPs [177]. As illustrated in Figure 9C, lectin could specifically recognize sialic acid which is over-expressed on the surface of cancer cells. The average amount of sialic acid on the surface of single cell has been evaluated by the biosensor.

Effective and robust nanoelectrocatalysts play an important role in the development of sensitive non-enzymatic electrochemical biosensors. Up to now, many nanomaterials have been reported to have the intriguing electrocatalytic activity for signal amplification. Noble metal NPs have attracted growing attention in exploring nanoelectrocatalysts for cytosensors [178,179]. Zhou et al. reported an amperometric immunosensor for CTCs detection with PtNPs as nanoelectrocatalysts through a tyramide-based signal ampliiation (TSA) system [180]. As displayed in Figure 10A, PtNPs were functionalized with nucleolin-targeting aptamer AS1411 (CP) and HRP (PtNPs@HRP@CP). Electroactive infinite coordination polymer (ICPs) self-assembled from ferrocenedicarboxylic acid molecules under sunshine were conjugated with tyramine (Tyr) (ICPs@Tyr). After HeLa cells were captured by CPs on the electrode, PtNPs@HRP@CPs were added to interact with the nucleolin on cell membrane. Under the PtNPs@HRP@CPs-catalyzed reaction of H_2_O_2_, a large amount of ICPs@Tyr polymers were deposited on the cell surface. Through the catalytic reaction of H_2_O_2_, the reduction peak current of ICP was significantly increased. Moreover, the multimetallic NPs always possess higher catalytic activity and stability than the monometallic NPs. Thus, many nanoalloys have been prepared and applied to develop sensitive cytosensors, including Au@Ag NPs, Pt@Ag nanoflowers, polyhedral Au@Pd NPs and mesoporous PdIrBP nanospheres [181,182,183,184,185,186]. Typically, Ge et al. reported an ultrasensitive electrochemical cytosensor based on trimetallic dendritic Au@PtPdNPs with a lab-on-paper device [187]. The robust electrocatalytic activity of Au@PtPdNPs resulted from the synergistic interaction of well-defined gold core and dendritic bimetallic shell. Meanwhile, the dendritic nanostructure possessed the improved catalytic capability of Au@PtPdNPs due to its enlarged surface area. Many studies indicated that nanohybrid of various NPs can further improve the electrochemical catalytic ability. Cu_2_O cubes with a large surface/volume ratio were used to load Pt@Pd NPs for CTCs detection (Figure 10B) [188]. Owing to the synergetic effect, the formed nanocomposite exhibited enhanced catalytic activity as the signal amplification label for CTCs detection. Con A was immobilized on the nanocomposite to identify apoptotic cells. Moreover, alloy NPs can be labeled with catalytic DNAzyme and HRP for multiple signal amplification [189,190]. For example, Sun et al. reported an electrochemical cytosensor using hybrid nanoelectrocatalysts and HRP for signal amplification [191]. As displayed in Figure 10C, the nanoelectrocatalysts were constructed from Fe_3_O_4_/MnO_2_ and Au@PdNPs by a layer-by-layer technique. It was further modified with HRP and an integrated DNA strand consisted of an aptamer sequence and a G-quadruplex-forming sequence. After the sandwich-like interaction, hybrid Fe_3_O_4_/MnO_2_/Au@Pd nanoelectrocatalysts, G-quadruplex/hemin HRP-mimicking DNAzymes and HRP efficiently catalyzed the oxidation of hydroquinone (HQ) with H_2_O_2_, amplifying the electrochemical signal and improving the detection sensitivity. This cytosensor achieved a wide detection range of 10^2^~10^7^ cells/mL with a detection limit of 15 cells/mL.

Fe_3_O_4_ NPs with intrinsic enzyme-mimicking activity were discovered in 2007 and have been commonly utilized to catalyze the electrochemical reduction of H_2_O_2_ [192]. Nevertheless, Zheng et al. found that Fe_3_O_4_ NPs can catalyze the electrochemical reduction of dyes without H_2_O_2_ [193]. As shown in Figure 10D, Ag-Pd bimetallic nanocages were assembled on the surface of Fe_3_O_4_ (Fe_3_O_4_@Ag-Pd) to enhance the catalytic signal for CTCs detection. After the cells were captured by the SYL3C aptamer-modified electrode, the aptamer-labeled Fe_3_O_4_@Ag-Pd was anchored on the cell surface to catalyze the reduction of thionine, thus producing a sharply increased current signal. The method showed a linear range of 50~10^7^ cells/mL. Tian et al. for the first time developed a CuO nanozyme-based electrochemical cytosensor for the detection of CTCs in breast cancer [194]. The CuO nanozyme could catalyze the decomposition of H_2_O_2_, thus producing an amplified signal. GO-based hybrid nanomaterials have been reported to possess peroxidase-mimicking ability. Liu et al. proposed an electrochemical cytosensor by using prickly Au nanoflowers to enhance the peroxidase-like property of GO-hemin-composite [195]. Based on the high electron transfer rate of GO and the synergistic interaction of the three components, K562 cells were sensitively detected in the range of 0–1 × 10^7^ cell/mL with a LOD of 10 cell/mL.

**Figure 10 biosensors-11-00281-f010:**
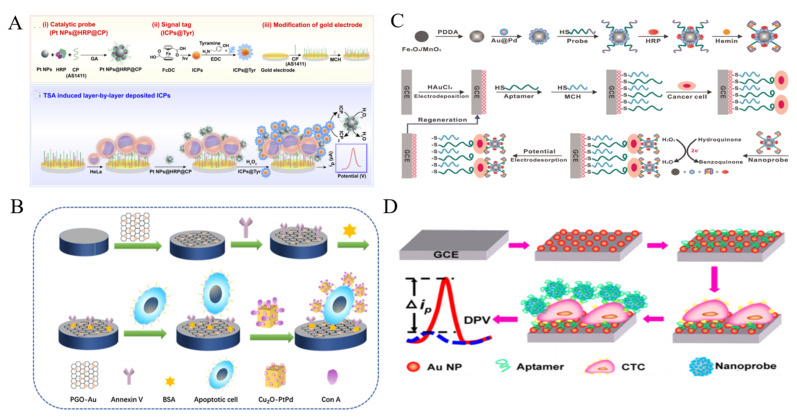
(**A**) Schematic representation of electrochemical immunosensor for CTCs (HeLa cell as a model) detection based on TSA induced layer-by-layer deposited ICPs [180]. Copyright 2019 Elsevier B.V. (**B**) Schematic representation of construction process of sandwich electrochemical cytosensor based on Cu_2_O cubes-Pt@Pd NPs [188]. Copyright 2019 Elsevier B.V. (**C**) Schematic representation of the fabrication of Fe_3_O_4_/MnO_2_/Au@Pd–HRP–aptamer/hemin/G-quadruplex hybrid nanoprobe and the EC cytosensor fabrication process [191]. Copyright 2016 Elsevier B.V. (**D**) Schematic representation of the cytosensor assembly process based on Fe_3_O_4_@Ag−Pd [193]. Copyright 2008 American Chemical Society.

Nanomaterials with electroactive properties have been widely applied in the field of biosensing and immunoassay with improving sensitivity. For example, reduced graphene oxide-tetrasodium 1,3,6,8-pyrenetetrasulfonic acid/metal hexacyanoferrates (rGO-TPA/MHCFnano) nanocomposites were used as the electroactive tags for the determination of SKBR-3 breast cancer cells [196]. Moreover, the in-situ generated electroactive materials can be introduced into the development of sensitive cytosensors [197]. Although QDs exhibit no electroactive properties, metal ions with sharp and well-resolved stripping voltammetric signals can released from QDs after the acid dissolution. Thus, QDs have been widely used as the electroactive nanotags in electrochemical biosensing. For example, Li et al. proposed a sensitive QDs-based electrochemical immunoassay to detect cancer cells with dual recognition elements [198]. As shown in Figure 11A, the MCF-7 cells were captured by the MUC1 aptamer-modified Au electrode and then anti-CEA-labeled CdS QDs were attached on the electrode surface. After the acid-dissolution, numerous Cd^2+^ ions were released into the solution and detected by anodic stripping voltammetry. FA-labeled CdSe/ZnS QDs were utilized to detect KB cells with over-expressed FRs [199]. Moreover, electroactive Cd^2+^-exchanged titanium phosphate nanospheres (TiP NSs) were synthesized and used to detect HL-60 cells [200]. Large quantities of Cd^2+^ in TiP NSs could directly produce a well-defined and sharp peak via in situ redox reaction without acid-dissolution.

Nanomaterials can be utilized as the nanocarriers to increase the loading amount of QDs in the signal nanoprobes [201]. For example, silica NPs and polystyrene microspheres were used to load a plenty of QDs via layer-by-layer technique for CTCs detection [202,203]. This multilabeled QDs-based signal amplification strategy dramatically increased the intensity of electrochemical signal and improved the sensitivity. Moreover, Wu et al. proposed an immunosensing method for CTCs detection by simultaneously measuring two different types of protein biomarkers on the surface of tumor cells [204]. As illustrated in Figure 11B, CdTe and ZnSe-coated silica NPs were modified with two different antibodies (anti-EpCAM and anti-GPC3). The electrode was decorated with electrochemically reduced GO to accelerate the electron transfer. After a sandwich-type immunoreaction, two types of QDs were bound to the cell surface and then dissolved by acid for square-wave voltammetric (SWV) stripping assays. Furthermore, long DNA concatamers formed through DNA hybridization can also be utilized to load plenty of QDs as the signal-amplified probes [205]. Liu et al. proposed a signal amplification supersandwich strategy for cancer cell detection based on the aptamer-DNA concatemer-QDs probes (Figure 11C) [206]. The sensing system consisted of an aptamer for specific recognition and QDs for signal amplification. Polydopamine (PDA) and AuNPs-decorated MWCNTs were deposited on the electrode surface to immobilize Con A and amplify the signal. After the immunoreaction, the released metal ions from the bound QDs were determined through a dual-channel route with fluorescent and electrochemical methods.

**Figure 11 biosensors-11-00281-f011:**
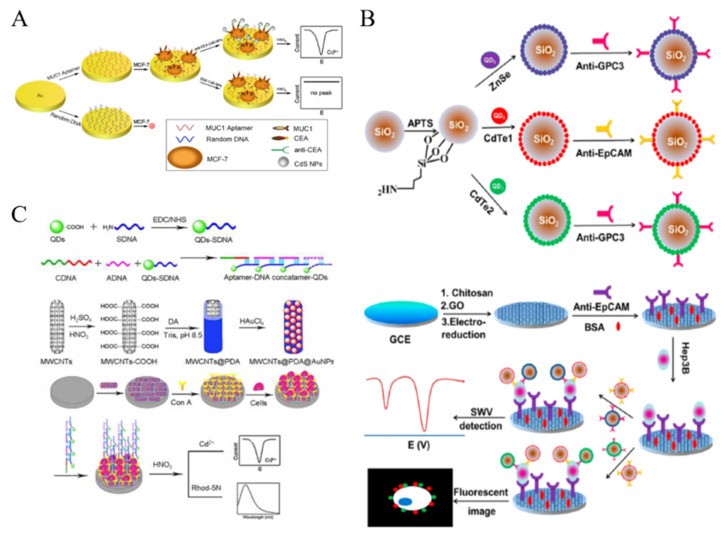
(**A**) Schematic representation of the method to detect breast cancer cells through simultaneous recognition of two different tumor markers [198]. Copyright 2010 Elsevier B.V. (**B**) Schematic representation of the preparation of antibody-modified QDs-loaded Si NPs and the detection procedure of CTCs [204]. Copyright 2013 American Chemical Society. (**C**) Schematic representation of the fabrication of aptamer-DNA concatamer-QDs and MWCNTs@PDA@AuNPs composites and super-sandwich cytosensor [206]. Copyright 2013 American Chemical Society.

In spite of low stability in saline buffer, AgNPs with extraordinary electrochemical properties have been widely used as the electroactive labels. It can generate a well-defined and sharp oxidation peak through the highly characteristic solid-state Ag/AgCl process. Thus, aptamer or peptide-labeled AgNPs have been used to detect tumor cells [207,208,209]. Tang et al. synthesized multifunctional nanofibers based on self-assembled peptide to develop a AgNP-based electrochemical cytosensor [210]. As shown in Figure 12A, peptide consisted of three parts, an azide group at the *N*-terminal for ligation with DBCO-AgNPs, a KLVFF motif for the formation of β-sheet structured nanofibers and a CD44 binding peptide (CD44BP) motif for selective recognition. The self-assembled nanofiber has numerous N_3_ groups for modification and signal amplification. After breast cancer stem cells (BCSCs) were captured by AS1411 aptamer, nanofibers were introduced to identify the cells and then DBCO-AgNPs were attached to the electrode surface by click chemistry. BCSCs can be effectively determined by strong silver response with a detection limit of 6 cells/mL. Furthermore, *p*-sulfonatocalix[4]-arene (*p*SC_4_)-modified AgNPs were utilized to label CTCs by binding to the multiple amino acid residues on the cell surface [211]. Compared to glycoyl residues for Con A recognition, the amino acid residues are relatively stable and can distinguish different types of cancers. Based on the high signal-to-noise ratio and low-potential electrochemical signal, *p*SC_4_-modified AgNP could be a universal electrochemical nanoprobe for cell analysis. For multi-marker analysis, Wan et al. developed a highly specific electrochemical method for CTCs detection by using multi-nanoparticle labels [212]. As illustrated in Figure 12B, a microfabricated chip with multiple anti-EpCAM aptamer-modified Au sensors was employed to capture cancer cells. Three different metal NPs, including Cu, Pd and Ag, were modified with aptamers and antibodies, respectively. After the capture and labeling of cells, the direct electrochemical oxidation of these metal NPs was conducted for the specific detection and characterization of cancer cells. Ag nanoclusters (AgNCs) can be facilely synthesized with DNA as the template. Meanwhile, the DNA template can be integrated with an extra sequence for specific recognition toward biomolecules or metal ions. Based on this fact, Li et al. presented an electrochemical cytosensor based on the reduction of disulfide bonds within membrane proteins in a mild condition and the binding of DNA bridge complex-templated AgNCs through the thiol-maleimide reaction [213]. As displayed in Figure 12C, anti-GPC3 antibody was immobilized on the pSC_4_-modified gold electrode through host-guest recognition. Target GPC3-positive HepG2 cells were treated with a mild reducing agent tris(2-carboxyethyl)phosphine (TCEP) to generate plentiful thiol sites and were captured by the electrode. Then, maleimide-labeled DNA was introduced to modify the cell surface through thiol-maleimide conjugation, which linked the DNA-templated AgNCs to the cell for signal amplification. NPs or enzyme-catalyzed silver enhancement is an effective strategy for signal amplification, which has been widely utilized in optical and electrochemical bioassays [214,215,216]. Guo et al. developed an electrochemical immunosensor for the detection of MCF-7 cells by employing MUC1 aptamer-containing DNA as the template for the synthesis of AgNCs [217]. In this work, AgNCs on the cell surface could mediate the metal silver deposition process and significantly improve the detection sensitivity.

#### 2.5.3. Other Methods

An aptamer-based competition/displacement strategy has been widely introduced into cytosensors based on the strong affinity between aptamers and target proteins on cell surface. In such a method, detection of cell concentration was converted into the assay of complementary DNA, which could be detected by various powerful DNA-based signal amplification strategies, such as catalytic hairpin assembly (CHA) and free-running DNA walkers [218,219]. Lu et al. reported a signal-amplified electrochemical aptasensor for cell detection based on supersandwich G-quadruplex DNAzyme [220]. As illustrated in Figure 13, aptamers attached on the surface of Fe_3_O_4_ NPs hybridized with cDNA. Then, the aptamer–cell interaction induced the release of cDNAs, which was proportional to the concentration of K562 cells. Then, the cDNAs triggered a cascade of hybridization events and resulted in the formation of the supersandwich structure. Numerous sequences suspended on the complexes were transformed into G-quadruplex DNAzymes in the presence of hemin, thus catalyzing the reduction of H_2_O_2_ by TMB and leading to the enhancement of peak current.

### 2.6. Electrochemiluminescence Cytosensors

Electrochemiluminescence (ECL) is one of the most important analytical techniques in bioassays [221,222,223]. Normally, the ECL detection system includes the electrochemical excitation and light emission without the use of additional excitation source. Thus, the method can avoid the interference of auto-fluorescence and scattered light. Up to date, many small molecules have been applied to design ECL biosensors for cell detection (e.g., luminal and tris(2,20-bipyridine)ruthenium(II) or Ru(bpy)_3_^2+^) [224,225,226]. Liu et al. reported that the positively charged Ru(bpy)_3_^2+^-modified indium tin oxide (ITO) electrode can capture the negatively charged Jurkat cells through the electrostatic interaction, thus achieving the detection of CTCs with a “signal-off” mode [227].

Nanomaterials can be utilized as the nanocarriers to load numerous ECL molecules for signal amplification [228,229]. For instance, Ru(II) molecules were covalently conjugated on mesoporous silica NPs (MSN) as the ECL tags for CTCs detection [230]. Based on the principle, one target cell can be converted into multiple ECL Ru(II) molecules, resulting in the enhancement of detection sensitivity. Zhou et al. developed an ECL cytosensor based on Iridium complex-encapsulated MSNs [231]. As displayed in Figure 14A, MSNs were used to load large amounts of Iridium complex. AuNPs were used to cap the mesopores and attach Con A molecules for recognizing the mannose groups on the cell surface. After the formation of the sandwich-like complexes on the electrode, a strong ECL signal was observed with tripropylamine (TPA) as the coreactant. Natural enzymes can improve the sensitivity through enzymatic amplification. For example, ALP molecules were loaded on AuNPs as labels for the development of ECL cytosensor by catalyzing the production of phenols that could hamper the ECL reaction of Ru(bpy)_3_^2+^ on the electrode interface [232].

Noble metal nanomaterials can also amplify the ECL intensity due to their unique merits of excellent electrocatalytic activity, large surface area and good electron conductivity [233,234]. β-Cyclodextrin-AuNPs were used to load Ru(bpy)_3_^2+^ for the development of a “signal-on” switch ECL biosensor for the detection of MEAR cells [235]. Ge et al. reported an ECL method for the detection of K563 cancer cells based on Au nanocages/Ru(bpy)_3_^2+^ molecules [236]. Hollow Au nanocages with porous structure and high surface area not only possessed more active sites to load a large amount of Ru(bpy)_3_^2+^ molecules, but also accelerate the electron communication. The proposed cytosensor exhibited a wide linear range of 500–5 × 10^6^ cell/mL. The bio-bar-code techniques integrated with MBs and different DNA-based amplification strategies were developed for sensitive detection of cells, including strand displacement reaction (SDR) and rolling circle amplification(RCA) [237]. Ding et al. reported the ECL assays of CTCs by employing Ru(II) complex-functionalized aptamers, AuNPs and MBs [238]. As shown in Figure 14B, bio-bar-code AuNPs were first modified with tris(2,2′-bipyridyl)ruthenium (TBR)-labeled signal reporter and linker as the ECL nanoprobes. The nanoprobes could be captured by the aptamer-conjugated MBs-1 via hybridization. The binding of target Ramos cells with the aptamers induced the release of AuNPs from the surface of MBs-1 into the solution. The released AuNPs were then captured by MBs-2 and deposited on the magnetic electrode for ECL detection. Recently, Chen et al. developed a PtNi anocbes-catalyzed TSA-based ECL cytosensor for the detection of hepatocellular carcinoma cells [239]. As illustrated in Figure 14C, after the formation of aptamer-cell-aptamer sandwich complex on the electrode, PtNi nanocbes could catalyze the covalently binding of numerous tyramine-luminol (Tyr-Lum) on the cell surface in the presence of H_2_O_2_. Then, nanocubes further promoted the Lum-H_2_O_2_ ECL system to generate a greatly enhanced ECL signal. On the basis of TSA and bifunctional PtNi nanocubes, the proposed cytosensor showed a wide linear range of 1 × 10^2^–1 × 10^7^ cell/mL and a LOD of 3 cell/mL.

**Figure 14 biosensors-11-00281-f014:**
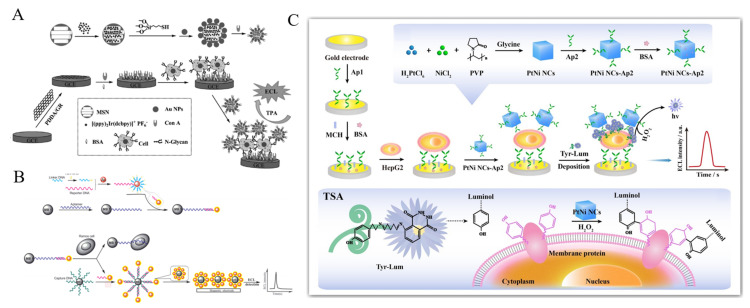
(**A**) Schematic representation of fabrication of the Con A@Au/Ir-MSN nanoprobe, and assembly of the electrode interface and the sandwich-like nanoarchitectured electrode geometry for cytosensing [231]. Copyright 2014 Wiley-VCH. (**B**) Schematic representation of the ECL cytosensor based on Ru(II) complex-functionalized aptamers, AuNPs and MBs [238]. Copyright 2012 Wiley-VCH. (**C**) Schematic representation of the prepared ECL cytosensor for HepG2 cells detection based on TSA strategy [239]. Copyright 2020 Elsevier B.V.

During the past decades, several types of nanomaterials have been exploited with good ECL property. For example, QDs have been used as the ECL emitters since they can be excited by electrical injection of an electron-hole pair to generate strong ECL signal [240]. For this view, Jiang et al. reported a label-free method for the assays of folate receptor (+) HeLa cells by molecular recognition (Figure 15A) [241]. In this work, the red emitting CdTe/GSH QDs were immobilized on the ITO electrode and then modified with FA. In the absence of cells, a strong ECL signal was observed with O_2_ as the co-reactant. Once folate receptor (+) HeLa cells were captured by the interaction between FA and folate receptor, the electron exchange between QDs and O_2_ was blocked, resulting in the decrease of ECL signal. The developed platform could detect ∼35 cells from 10 μL of cell suspension. To enhance the sensitivity of ECL methods, several interesting strategies have been proposed by employing the nanomaterials as the ECL enhancers, the layer-by-layer technique for increasing the dose of QDs on the electrode surface, and the nanohybrids for signal enhancement. For example, MBs were utilized to carry CdS QDs for the development of ECL cytosensor based on the interaction between the epidermal growth factor and the epidermal growth factor receptor on cell surface [242]. Highly branched dendrimer can be used to load large numbers of QDs for improving the density of QDs and enhancing the ECL signal [243]. Jie et al. first reported a versatile ECL method for CTCs detection based on dendrimer/CdSe-ZnS QDs nanoclusters [244]. As shown in Figure 15B, the fifth-generation PAMAM dendrimers with abundant amine groups were used to load QDs and aptamers, which greatly amplified the ECL signal. After the magnetic separation, target cells induced the release of intermediate DNA strands in solution. This initiated a nicking endonuclease-assisted cycle-amplification on MBs and plenty of QDs were liberated into the solution. The released QDs can be recruited on the electrode surface to produce a strong ECL signal. Moreover, AuNPs were also employed to quench the ECL signal of dendrimer/CdSe-ZnS QDs nanoclusters, and the target cells could liberate QDs, resulting in the recovery of ECL signal [245]. To simplify the experiment procedures, Jie et al. synthesized magnetic QDs and used them to detect cancer cells by endonuclease-assisted cyclic amplification [246]. Nanoparticles with a large surface area are beneficial to the improvement of ECL signal by accelerating the electron transfer [247,248]. The nanocomposite of TiO_2_ and CdS QDs were utilized to detect HepG2 cells and the ECL signal was enhanced by ionic liquid and AuNPs [249]. Ionic liquid facilitated the immobilization of the nanocomposite on the electrode and improved the electron transfer from NPs to the electrode.

AuNPs can play two different roles (quencher or enhancer) in QDs-based ECL system, which is dependent on the distance between QDs and AuNPs. For example, Zhang et al. reported a AuNPs-enhanced ECL method for cancer cell detection (Figure 15C) [250]. Glass carbon electrode (GCE) was modified with CdS nanocomposite film and DNA aptamer. After the specific capture of HL-60 cancer cells by the aptamer-conjugated MBs, AuNPs were released and then captured by the QDs-modified GCE. The enhancement of ECL intensity was observed due to the overlap between electrogenerated excitons of QDs and LSPR of AuNPs. Recently, ratiometric ECL biosensors have been developed for sensitive bioassays without the interferences of external factors. Ding et al. reported a ratiometric ECL cytosensor using CdTe QDs as the detection probes and luminol molecules as the internal standards [251]. As shown in Figure 15D, polyaniline-based conducting polymer hydrogel (CPH) film was electrodeposited on the electrode surface and luminol and coreactant potassium persulfate (K_2_S_2_O_8_) were encapsulated into the hydrogel. The MUC1 aptamer and nucleolin aptamer were used to modify AuNPs and CdTe QDs for improving the recognition efficiency, respectively. The dual-aptamer sensor showed high selectivity and sensitivity in contrast to the single-aptamer sensor.

Carbon dots (CDs) have been widely utilized as the novel ECL luminophores because of their remarkable luminous efficiency, excellent stability and good biocompatibility. Liu et al. reported the label-free ECL detection of CTCs using gold@CDs nanohybrids (Au@CDs) [252]. As displayed in Figure 16A, Au@CDs were synthesized using CDs as the reducing and capping agents to reduce chloroauric in-situ into AuNPs. The nanohybrids showed 12-fold higher ECL emission than that of CDs alone, which could be used to modify the electrode for direct detection of cancer cells. After the capture of cells by MUC1 aptamer-modified electrode, cells hindered the ECL reaction between the hybrid and the co-reactant, resulting in the quenching of ECL signal. In addition, Su et al. developed an ECL platform for the detection of MCF-7 cancer cells by using CQDs-coated silica NPs as ECL tracers [253].

Graphite carbon nitride (g-C_3_N_4_) nanosheets composed of carbon and nitrogen atoms have a typical two-dimensional (2D) graphite-like structure and a moderate band gap. It shows excellent fluorescence and ECL properties and has great potential in the field of biosensing for CTCs [254]. He et al. reported a reusable and dual-potential response ECL cytosensor using Ru(phen)_3_^2+^ and AuNPs-modified g-C_3_N_4_ (Au-C_3_N_4_) as the ECL probes at positive and negative potentials, respectively [255]. As shown in Figure 16B, aptamers on the electrode surface hybridized with the capture probes of DNA and the hybrids were intercalated with Ru(phen)_3_^2+^ molecules. Target cells specifically bound to the aptamers and broken the complexes, inducing the release of Ru(phen)_3_^2+^ molecules from the electrode interface. Then, ConA-modified Au-C_3_N_4_ was used to label the cells and served as a negative ECL nanoprobe. MCF-7 cancer cells were sensitively determined based on the ratio of ECL intensity between the negative and positive potential. Feng et al. proposed a ratiometric ECL method for the detection of CTCs and cell-surface glycans (Figure 16C) [256]. g-C_3_N_4_ nanosheets as the cathodic nanoemitters were modified with AuNPs and aptamers. Luminol-loaded AuNPs (LuAuNPs) as the anodic nanoemitters were conjugated with lectin to recognize the glycan groups on cell surface. After the addition of cells, the anodic ECL signal of LuAuNPs increased and the cathodic ECL intensity of g-C_3_N_4_ decreased, because of the blockage of cells and the compete consumption of co-reactant H_2_O_2_. The ratio of anodic to cathodic ECL intensity showed good linear relativity with the concentration in the range of 10^2^ ~ 10^6^ cells/mL.

The morphology and surface vacancies of ECL materials have an important effect on the performances of analytical methods. Recently, Gao et al. synthesized the 2D ultrathin Lu_2_O_3_-S nanosheets with abundant oxygen vacancies for developing ECL cytosensor by coupling with enzyme-assisted DNA amplification [257]. As shown in Figure 16D, Lu_2_O_3_-S nanosheets exhibited an enhanced ECL signal and the DNA-modified Ag_2_S QDs could quench the luminescence through the ECL-RET effect. Through hybridization-competition strategy, the target CCRF-CEM cells were transformed into programmable DNA, which further initiated the T7 exonuclease-assisted cycle-amplification. The low abundance of cells could induce a lot of Ag_2_S QDs to leave away from the electrode surface, thus leading to the recovery the ECL signal. The proposed ECL cytosensing exhibited a linear concentration range of 10~10^6^ cells/mL with a detection limit of 10 cells/mL.

**Figure 16 biosensors-11-00281-f016:**
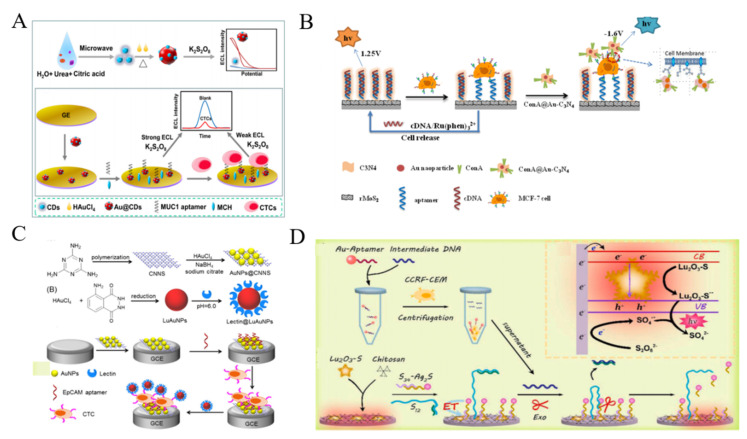
(**A**) Schematic representation of synthesis protocol and structural model of Au@CDs and principle of the Au@CDs-based ECL cytosensor for CTCs detection [252]. Copyright 2020 Elsevier B.V. (**B**) Schematic illustration of ECL biosensor for carbohydrate expression on cell surface based on ConA-modified Au-C_3_N_4_ [255]. Copyright 2015 American Chemical Society. (**C**) Schematic representation of fabrication of cathodic probe AuNPs@CNNS and anodic probe lectin@LuAuNPs, and dual-potential ratiometric ECL strategy for detection ofCTCs [256]. Copyright 2016 Elsevier B.V. (**D**) Schematic representation of the Lu_2_O_3_-S nanosheets-based EL cytosensor [257]. Copyright 2021 Elsevier B.V.

### 2.7. Photoelectrochemical Cytosensors

Photoelectrochemical (PEC) process mainly involves in the charge transfer between the photoactive materials and the electrolyte under light irradiation. In this process, light energy is absorbed by the photoactive substrate and then transformed into electricity. In addition to the inherited merits of electrochemical bioanalysis, the independent energy forms of light (input signal) and current (output signal) endow PEC technique with the advantage of low background noise signal. The technique has witnessed rapid development in past decades and has been intensively utilized in bioanalysis, including aptasensing, immunoassay and cytosensing [258,259]. In this section, the related works were summarized according to the roles of nanomaterials in PEC assays, acting as photoelectrode materials to modify electrode and acting as the PEC signal modulators.

As an indispensable component in a typical PEC sensor, photoactive materials have been rationally designed and engineered into the electrode interface, including QDs, metal oxides metal chalcogenides and so on. Generally, BEs are used to modify the surface of nanomaterials-decorated electrode to capture CTCs. Due to the steric hindrance effect, the captured CTCs can change the diffusion of electrolyte to the surface of photoactive nanomaterials and impede the reaction. Thus, CTCs could be directly determined without extra labeling steps. Many PEC cytosensors have been reported based on this principle [260]. Semiconductor nanostructures, including QDs and metal oxide are the most popular photoactive nanomaterials in the development of PEC biosensors [261,262]. Zhang et al. proposed an aptamer-based label-free PEC cytosensor with photoactive films formed by the layer-by-layer assembly of CdSe QDs [263]. A large amount of photoelectrochemically active QDs on the electrode could produce high photocurrent and increase the sensitivity of biosensor. However, when the aptamer-CTC complex was formed, the diffusion of AA to the electrode was blocked, resulting in the decrease of photocurrent. Moreover, Xu et al. constructed a renewable PEC cytosensing platform for fast capture and detection of MCF-7 cells [264]. As shown in Figure 17A, CdS/ZnO hybrid nanorods were grown on the ITO slides and decorated with layer of polymerized aminophenylboronic acid (APBA). Abundant phenylboronic acid groups in the film could specifically interact with sialic acid over-expressed on the membrane of CTCs through the formation of boronate ester bonds. Due to the reversibly of boronate ester bonds, the captured cells could be released. The proposed cytosensor exhibited a broad linear range and a low detection limit of 10 cells/mL. However, the wide band gap and photocorrosion effect limited their practical applications.

Biocompatible hexagonal carbon nitride tubes (HCNTs) can improve light harvesting and enhance charge carrier transfer. HCNTs were utilized as the photoactive materials to develop an aptamer-based PEC method for the determination of MCF-7 cells [265]. After the binding of cells onto HCNTs, the photocurrent intensity was suppressed due to the steric hindrance effect. The AuNPs/g-C_3_N_4_ nanocomposites were employed to construct a light-driven self-powered sensing system for the detection of CTCs through the integration of biofuel cells and PEC technique [266]. Glucose was directly oxidized by the photogenerated holes without the introduction of anodic enzymes. In 2016, Wang et al. reported the first NIR-driven PEC cytosensor by using upconversion NaYF_4_:Yb,ErNPs and TiO_2_/CdTe heterostructure [267]. As illustrated in Figure 17B, the upconversion NPs could convert the NIR light (900 nm) to visible light, which was re-used by CdTe/TiO_2_ to generate the photogenerated electrons. The captured MCF-7 cells blocked the charge transfer and reduced the concentration of sacrificial agents near the ITO electrode surface, resulting in the decrease of the photocurrent response.

Similar to the sandwich-like detection mode of electrochemical method, nanomaterials can also be used to label the captured cells on the electrode by modification with BEs. Then, the photocurrent signal was modulated by nanomaterials or components on the electrode surface. Competitive consumption of the electrolyte or absorption of the exciting light are another effective strategy to develop PEC biosensors. As a *p*-type semiconductor, cuprous oxide (Cu_2_O) with a narrow band gap of about 2.0 eV shows a great potential in PEC application. Luo et al. proposed a PEC method for CTCs detection by using aptamer-modified Cu_2_O [268]. As shown in Figure 17C, HCNTs were utilized as photoactive materials to modify the electrode. After magnetic separation, the MNs-CTCs complex was deposited onto the electrode, resulting in the decrease of photocurrent intensity due to the steric hindrance. Then, aptamer-Cu_2_O nanoprobe was linked to the captured cells and Cu_2_O NPs competitively absorb the light, leading to the sharply decreased photocurrent intensity and amplifying the PEC response. Ge et al. reported an ultrasensitive paper-based PEC cytosensor based on the dual competitive strategy [269]. As displayed in Figure 17D, ZnO/CdTe/AuNRs-assembled MSNs (GMSNs) nanostructure was fabricated on the electrode. The intensive photocurrent was dramatically enhanced by the nanostructure. However, when CTCs were captured by aptamers and further labeled with Con A-modified grapheme QDs (GQDs), GQDs would competitively absorb the light and consume H_2_O_2_, resulting in the significant decrease of the photocurrent intensity. This method can determine MCF-7 cells and in-situ identify the *N*-glycan expression on cell surface.

**Figure 17 biosensors-11-00281-f017:**
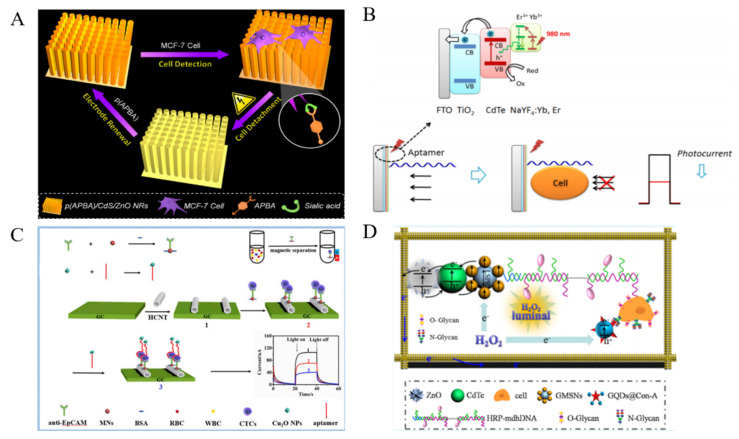
(**A**) Schematic representation of renewable PEC cytosensor for CTC capture and refresh [264]. Copyright 2021 Elsevier B.V. (**B**) Schematic representation of the NIR-light-excited PEC interface and the MCF-7 cell detection principle [267]. Copyright 2016 American Chemical Society. (**C**) Schematic representation of CTC measurement in whole blood based on MNs isolation of CTCs and Cu_2_O-aptamer as signal probe [268]. Copyright 2020 Elsevier B.V. (**D**) Schematic representation of paper-based PEC cytosensor based on the dual competitive strategy [269]. Copyright 2017 American Chemical Society.

**Table 2 biosensors-11-00281-t002:** Analytical performances of various nanomaterials-based electrochemical techniques for CTCs detection.

Detection Methods	Type of Nanomaterials	Detection Limit (Cells/mL)	Linear Range (Cells/mL)	Ref.
Electrochemistry (direct detection)	CNTs	10	10–1 × 10^5^	[133]
		10	10–1 × 10^6^	[135]
		90	2.7 × 10^2^–2.7 × 10^7^	[136]
		500	5 × 10^2^–5 × 10^6^	[137]
	GO	30	1 × 10^2^–1 × 10^7^	[140]
		5	5–1 × 10^5^	[141]
	AuNPs	6	6–1 × 10^3^	[143]
	Non-spherical AuNPs	2	5–2 × 10^6^	[142]
	Au nanoisland	30	1 × 10^2^–1 × 10^5^	[147]
	Ag@BSA microspheres	20	60–1.2 × 10^8^	[144]
	3D-structured microspheres assembled from CNSs and AuNPs	14	42–4.2 × 10^6^	[151]
	MnFePBA@AuNP	36	5 × 10^2^–5 × 10^4^	[152]
	NiCoPBA	47	1 × 10^2^–1 × 10^6^	[270]
	2D MoS_2_	0.43	1–1 × 10^5^	[271]
	TiO_2_ nanotubes@rGO	40	1 × 10^3^–1 × 10^7^	[272]
	MOFs	90	1 × 10^2^–1 × 10^6^	[154]
		19	1 × 10^2^–1 × 10^5^	[153]
	COFs	61	5 × 10^2^–1 × 10^5^	[155]
	Au NSs	5	5–1 × 10^5^	[148]
Electrochemistry (sandwich detection)	AuNPs	30	1 × 10^2^–1 × 10^7^	[165]
		1500	2 × 10^3^–2 × 10^6^	[167]
		10	1 × 10^2^–5 × 10^4^	[169]
		100	1 × 10^2^–1 × 10^3^	[174]
	Porous PtFe alloys	38	1 × 10^2^–5 × 10^7^	[176]
	Fe_3_O_4_ NPs@AuNPs	660	1 × 10^3^–1 × 10^6^	[168]
	ZnO@Au-Pd	10	1.0×10^2^–1.0×10^7^	[170]
	MOFs@AuNPs	5	1 × 10^2^–1 × 10^7^	[171]
	MOFs	6	20–1 × 10^7^	[173]
	Pt NPs@HRP ICP@Tyr	2	2–2 × 10^4^	[180]
	Au@Ag NPs	6	1–5 × 10^5^	[181]
	Pt@Ag NFs	3	20–1 × 10^6^	[182]
	Polyhedral-AuPd NPs	20	50–1 × 10^7^	[183]
	Pd@Au NPs	30	1 × 10^2^–2 × 10^6^	[184]
	Dendritic Au@PtPd NPs	31	1 × 10^2^–2 × 10^7^	[187]
	Cu_2_O@PtPd nanocomposites	20	50–5 × 10^7^	[188]
	Fe_3_O_4_ bead@Ag-Pd nanocages	34 for MCF-7; 42 for T47D	50–1 × 10^7^	[193]
	CuO NPs	27	50–7 × 10^3^	[194]
	CdS QDs	3.3×10^2^	1 × 10^4^–2 × 10^7^	[198]
	CdSe/ZnS QDs	2.0×10^3^	5 × 10^3^–5 × 10^5^	[199]
	CdTe QDs-labeled SiNSs	1.0×10^3^	1 × 10^3^–1 × 10^7^	[202]
	CdS QDs-decorated PS	3	10–1 × 10^7^	[203]
	QDs-coated SiNSs	Not reported	5–1 × 10^6^ for Hep3B; 10–1 × 10^6^ for BGC	[204]
	Aptamer-DNA concatamer-QDs	50	1 × 10^2^–1 × 10^6^	[206]
	Cd^2+^-functionalized TiP NSs	35	1 × 10^2^–1 × 10^7^	[200]
	AgNPs	25	1 × 10^2^–1 × 10^7^	[207]
		6	10–5 × 10^5^	[210]
		5	5–2.5 × 10^5^	[211]
	AuNPs and AuNP-enhanced silver deposition	10	1 × 10^2^–1 × 10^6^	[208]
Electrochemiluminescence	Ru(II) markers-loaded SiNPs	78	1 × 10^2^–2 × 10^3^	[230]
	Iridium complex-encapsulated SiNPs	100	1 × 10^2^–1 × 10^6^	[231]
	Ru(bpy)^2+^-loaded Au cage	500	5 × 10^2^–5 × 10^6^	[236]
	CdTe QDs	3.5×10^3^	3 × 10^3^–3.5 × 10^5^	[241]
	dendrimer/QDs nanoclusters	320	1.6 × 10^2^–1.536 × 10^4^	[243]
		68	1 × 10^2^–4 × 10^3^	[244]
	magnetic composite QDs	98	3 × 10^2^–9 × 10^3^	[246]
	Au@CDs	34	1 × 10^2^–1 × 10^4^	[252]
	CQDs-coated SiNPs	230	5 × 10^2^–2 × 10^7^	[253]
	Au-C_3_N_4_	150	1 × 10^2^–1 × 10^6^	[255]
	Lu_2_O_3_-S nanosheets	10	10–1 × 10^6^	[257]
Photoelectrochemistry	CdSe QDs	84	1.6 × 10^2^–1.6 × 10^3^	[263]
	AuNPs/g-C_3_N_4_	10	20–2.0 × 10^5^	[266]
	HCNTs	17	1 × 10^2^–1 × 10^5^	[265]
	CdS/ZnO hybrid nanorods	10	50–1 × 10^6^	[264]
	CPs	24	1 × 10^2^–5 × 10^5^	[258]

Abbreviates: CNTs, carbon nanotubes; GO, graphene oxide; GAM, graphene aerogel microspheres; AuNPs, gold nanoparticles; BSA, bovine serum protein; CNSs, carbon nanospheres; MnFePBA, MnFe Prussian blue analogue; rGO, reduced graphene oxide; MOFs, metal organic frameworks; FA, folic acid; Au NSs, gold nanostars; LSPR, localized surface plasmon resonance; HRP, horseradish peroxidase; ICP, infinite coordinate polymer; Tyr, tyramine; NFs, nanoflowers; QDs, quantum dots; SiNPs, silica nanospheres; PS, polystyrene microspheres; TiP NSs, titanium phosphate nanospheres; AgNPs, silver nanoparticles; CDs, carbon dots; CQDs, carbon quantum dots; HCNTs, hexagonal carbon nitride tubes; CPs, conjugated polymers.

### 2.8. Other Detection Techniques

Due to its good anti-interference ability and high sensitivity, inductively coupled plasma mass spectrometry (ICP-MS) can also be utilized to determine biomolecules (such as protein, DNA and cells) by combining with element labeling strategy [273]. Metal-containing NPs have been used in ICP-MS-based cytosensors due to the large amount of metal atoms per nanoparticle, such as AuNPs, UCNPs and semiconductor QDs [274,275,276,277,278]. Moreover, the utilization of NPs with different elements can realize the simultaneous detection of two types of cells by ICP-MS. For instance, as shown in Figure 18A, anti-ASGPR-labeled QDs and anti-MUC1-labeled AuNPs were used as signal probes for the detection of HepG2 and MCF-7 cells by the ICP-MS simultaneous measurement of ^111^Cd and ^197^Au [279]. For achieving higher sensitivity, DNA-based amplification strategy was introduced into ICP-MS-based bioassays [280].

During recent years, a plenty of novel detection methods by using temperature and pressure as signals have been proposed for bioassays. Zhang et al. reported a paper-based cytosensor for CTCs based on the photothermal effect of GO-functionalized magnetic microbeads (MMBs) [281]. As shown in Figure 18B, the immune-GO-MMB complexes were formed by the interaction between antibodies and anti-IgG antibodies. After the immunoreaction, cells labeled with the complexes were separated by membrane filtration and further purified through magnetic separation. Under the laser irradiation, GO on the cells surface increased the temperature, which was monitored by an infrared camera. This group also employed AuNPs to enhance the photo-thermal energy conversion efficiency for CTCs detection [282]. Furthermore, Hu et al. reported a thermal-controlled pressure-based cytosensing of CTCs by using NIR light-responsive hollow porous gold nanospheres (AuNSs) (Figure 18C) [283]. After the cells were labeled with aptamer-modified AuNSs, the heat released from NIR light-activated AuNSs accelerated the decomposition of NH_4_HCO_3_ into NH_3_, CO_2_ and H_2_O in a closed container. CTCs were indirectly determined by monitoring the change of gas pressure with a pressure meter.

**Figure 18 biosensors-11-00281-f018:**
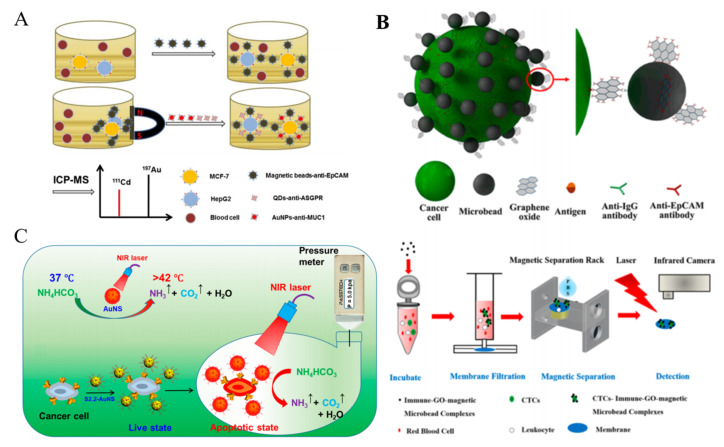
(**A**) Schematic representation of simultaneous detection of MCF-7 and HepG2 cells by ICP-MS with Au NPs and QDs as elemental tags [279]. Copyright 2017 Elsevier B.V. (**B**) Schematic representation of detection of CTCs based on the photothermal effect of GO-functionalized MMB [281]. Copyright 2016 American Chemical Society. (**C**) Schematic representation of the thermal-controlled pressure-based cytosensing of CTCs by using NIR light-responsive AuNSs [283]. Copyright 2019 American Chemical Society.

## 3. Conclusions

In this review, the current achievements of nanomaterials-based cytosensors for the detection of CTCs have been summarized. Advances in engineering functional nanomaterials and methodologies have significantly improved the sensitivity and reduced the detection limit. Moreover, the successful introduction of various elaborate signal amplification strategies based on DNA assembly and enzymes further enhances the analytical performances. Even though large amounts of works have been reported, there is still a big and crucial gap between academic research and clinical translation. For example, the widely used biorecognition elements (antibodies and aptamers) may suffer from rapid degradation in complex clinical samples that contains nucleases, proteases and other potential interferences. Even though pre-concentration can decrease the interference, it will increase the complexity of the detection procedure and may reduce the viability of cells. New systems and devices that permit cells to be analyzed in situ and recovered for further characterization are benefit to better extract more information about their phenotype and clinical relevance. Integrated artificial intelligence and emerging nanotechnology and powerful analytical methods may provide a promising approach. Furthermore, the stability and function of nanomaterials under physiological conditions should be considered. Proper surface modifications may resolve these challenges, but they may bring new problems. For instance, the active sites of nanozymes and nanoelectrocatalysts would be blocked due to the surface modification. Meanwhile, suitable modification methods should be explored to confirm the aptamer configuration and antibody orientation on the surface of nanomaterials and nanostructures because they can influence the affinity between the biorecognition elements and CTCs.

## Figures and Tables

**Figure 1 biosensors-11-00281-f001:**
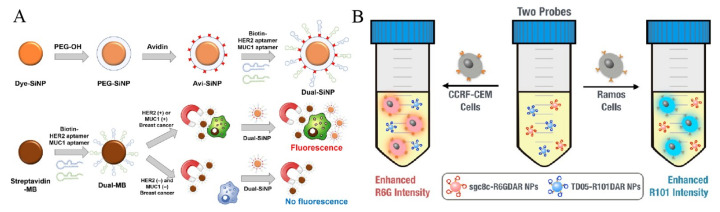
(**A**) Schematic representation of modification processes of dye-SiNPs and the selective detection for only MUC1(+) and HER2(+) breast cancer cells by Dual-SiNPs [68]. Copyright 2015 Elsevier B.V. (**B**) Schematic representation of cell detection/identification using Sgc8c-R6GDAR and TD05-R101DARNPs based on “turn-on” retro-self-quenched fluorescence [70]. Copyright 2015 American Chemical Society.

**Figure 2 biosensors-11-00281-f002:**
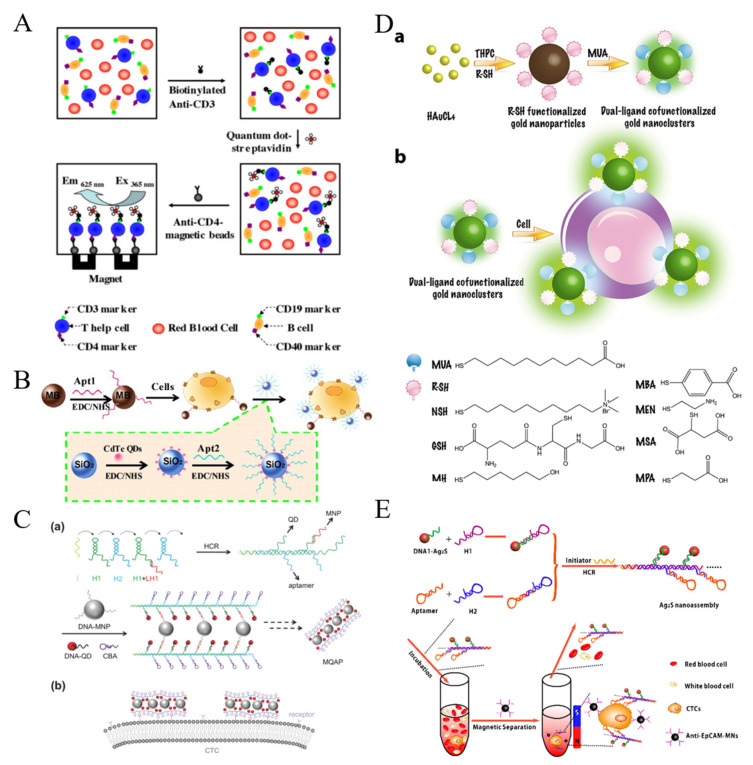
(**A**) Illustration of detecting the specific T-help cells using combination of two biomarkers with quantum dots as fluorescence probe and magnetic beads for immuno-separation [75]. Copyright 2011 Elsevier B.V. (**B**) Illustration of isolation and detection of MCF-7 cells by employing MUC1 aptamer-functionalized MBs and AS1411 modified QDs loaded on the surface of SiNPs [76]. Copyright 2013 Elsevier B.V. (**C**) Schematic illustration of MQAPs for magnetic isolation of CTCs. (a) Construction of MQAPs in two steps: (i) hybridization chain reaction mediated formation of the polymeric DNA template and (ii) co-assembly of DNA-QDs, DNA-MNPs, and DNA aptamers with the DNA template. (b) Multivalent binding between MQAPs and cell surface receptors for CTC isolation and detection [80]. Copyright 2018 WILEY-VCH. (**D**) Schematic illustration of the synthesis of the dual-ligand co-functionalized AuNCs and preparation of sensor array for cell identification based on the dual-ligand co-functionalized AuNCs. (a) Schematic illustration of the synthesis of the dual-ligand cofunctionalized gold nanoclusters. (b) Preparation of sensor array for cell identification based on dual-ligand cofunctionalized gold nanoclusters [81]. Copyright 2017 Elsevier B.V. (**E**) Schematic illustration of the procedure of isolation and detection of CTCs through the combination of Ag_2_S nanoassembly and anti-EpCAM-MNs [82]. Copyright 2018 American Chemical Society.

**Figure 3 biosensors-11-00281-f003:**
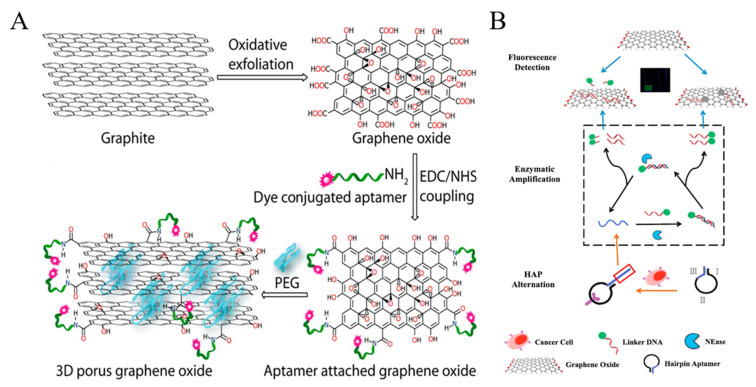
(**A**) Schematic representation showing the stepwise chemical formation of 3D graphene oxide [88]. Copyright 2015 American Chemical Society. (**B**) Schematic representation of a label-free and high-efficient GO-based aptasensor for the detection of low quantity cancer cells [91]. Copyright 2017 Elsevier B.V.

**Figure 4 biosensors-11-00281-f004:**
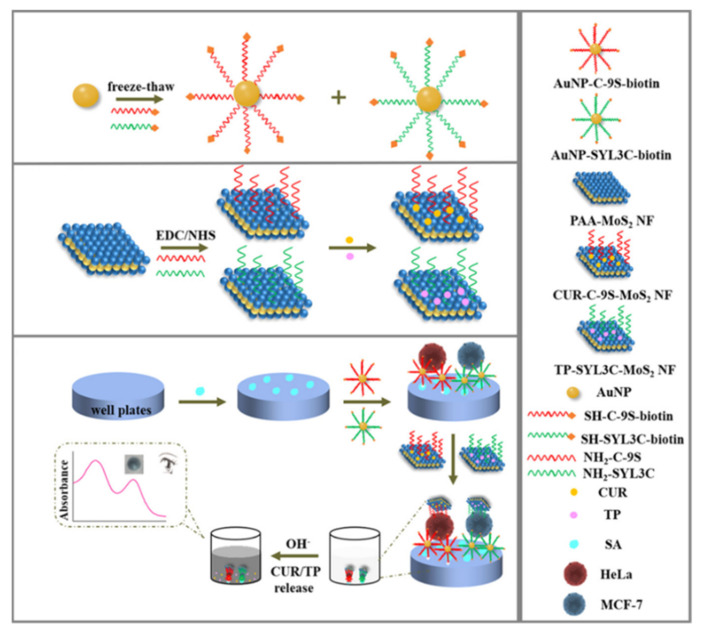
Illustration of pH-responsive nanotags-based nanobioplatform for the efficient capture and colorimetric detection of heterogeneous CTCs [96]. Copyright 2021 American Chemical Society.

**Figure 6 biosensors-11-00281-f006:**
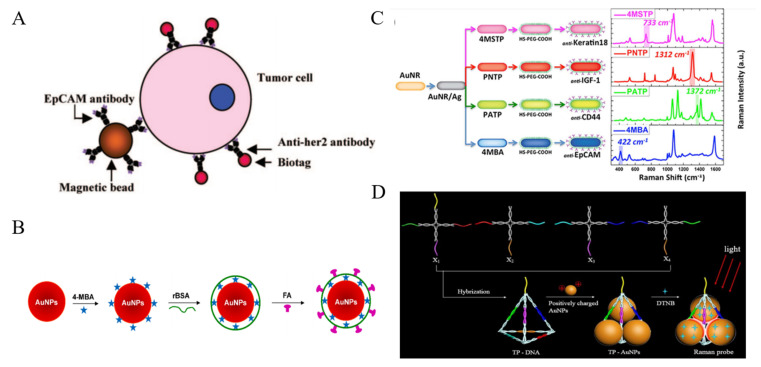
(**A**) Schematic representation of the ternary immuno-complex formed by nanoplexbiotags and MBs binding to the tumor cell [113]. Copyright 2008 American Chemical Society. (**B**) Schematic representation of the design of AuNPs-4-MBA-rBSA-FA SERS NPs [115]. Copyright 2015 American Chemical Society. (**C**) Schematic diagram (preparation steps) and Raman spectra (acquisition time 50 s) for the four families of SERS nano-agents [121]. Copyright 2014 Springer Nature. (**D**) Schematic representation of the synthesis of TP-AuNPs as the Raman probe [123]. Copyright 2019 American Chemical Society.

**Figure 9 biosensors-11-00281-f009:**
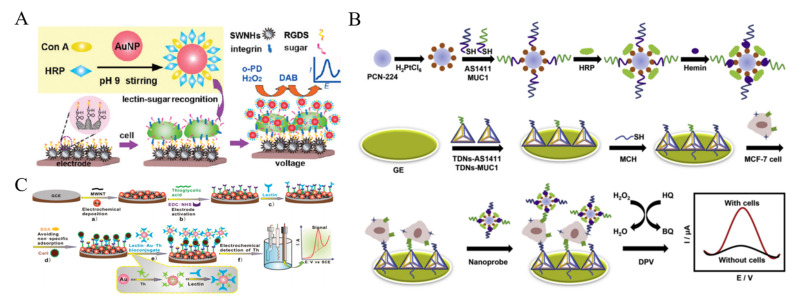
(**A**) Schematic representation of nanoprobe assembly and electrochemical strategy for in situ detection of cells and mannose groups on living cells [167]. Copyright 2010 American Chemical Society. (**B**) Schematic representation of fabrication procedures of PCN-224-Pt/HRP/dual-aptamer/GQH nanoprobe and the electrochemical dual-aptamer cytosensor fabrication process [173]. Copyright 2019 Elsevier B.V. (**C**) Schematic representation of the lectin-based biosensor for EC Analysis of cells [177]. Copyright 2010 American Chemical Society.

**Figure 12 biosensors-11-00281-f012:**
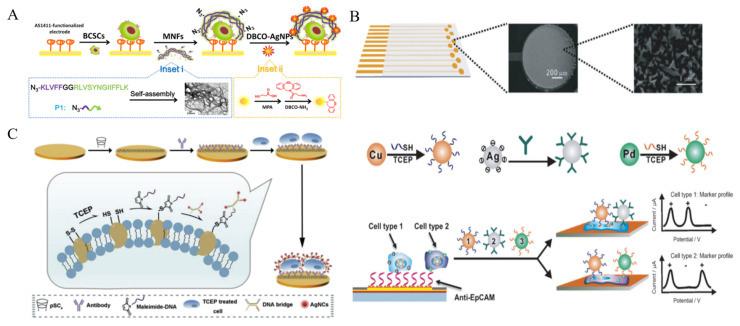
(**A**) Schematic representation of the multifunctional nanofiber-assisted EC identification of BCSCs [210]. Copyright 2019 American Chemical Society. (**B**) Schematic representation of the multi-nanoparticle approach to specific cancer cell detection [212]. Copyright 2014 Wiley-VCH. (**C**) Schematic representation of signal labeling and electrochemical sensing of cancer cells via mild reduction [213]. Copyright 2019 Elsevier B.V.

**Figure 13 biosensors-11-00281-f013:**
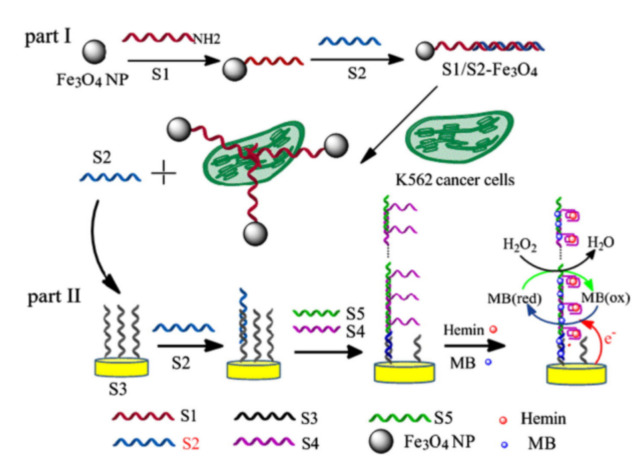
Schematic representation of the signal-amplified electrochemical aptasensor based on supersandwich G-quadruplex DNAzyme [220]. Copyright 2015 Elsevier B.V.

**Figure 15 biosensors-11-00281-f015:**
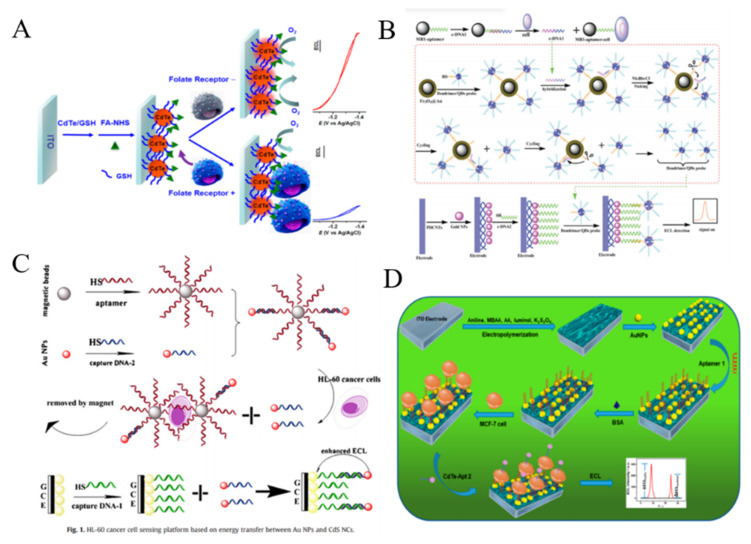
(**A**) Schematic representation of selective binding of FR (+) cells on FA functionalized CdTe/GSH NPs modified ITO electrodes [241]. Copyright 2014 American Chemical Society. (**B**) Schematic representation of the dendrimer NCs/QDs-DNA probe and ECL biosensor for signal-off detection of cells [244]. Copyright 2011 American Chemical Society. (**C**) Schematic representation of HL-60 cancer cell sensing platform based on energy transfer between Au NPs and CdS QDs [250]. Copyright 2012 Elsevier B.V. (**D**) Schematic representation of the ratiometric ECL cytosensor based on conducting polymer hydrogel loaded with internal standard molecules [251]. Copyright 2019 American Chemical Society.

## Data Availability

Not applicable.

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
