# Peer review of "Recent Development of Nanomaterials-Based Cytosensors for the Detection of Circulating Tumor Cells"

_biosensors, 2021, doi:10.3390/bios11080281_

Round 1

Reviewer 1 Report

1. Introduction: Is this the first review paper on the topic? Absolutely not. There have been many review papers on the detection of CTCs, which should be cited and critically reviewed in the manuscript. It is important to acknowledge the past review efforts and explicitly state what new perspectives the current paper brings to the community. From my quick search, I found the following review papers related to the nanomaterial-based CTC sensing just for Year 2020 and 2021. A lot more review papers can be found in the previous years.

[R1] Li, X.-R., Zhou, Y.-G., Electrochemical detection of circulating tumor cells: A mini review, Electrochemistry Communications 124 (2021)106949

[R2] Farshchi, F., Hasanzadeh, M., Microfluidic biosensing of circulating tumor cells (CTCs): Recent progress and challenges in efficient diagnosis of cancer, Biomedicine and Pharmacotherapy, 134 (2021) 111153

[R3] Koo, K.M., Soda, N., Shiddiky, M.J.A., Magnetic nanomaterial–based electrochemical biosensors for the detection of diverse circulating cancer biomarkers, Current Opinion in Electrochemistry, 25 (2021) 100645

[R4] Fattahi, Z., Khosroushahi, A.Y., Hasanzadeh, M., Recent progress on developing of plasmon biosensing of tumor biomarkers: Efficient method towards early stage recognition of cancer, Biomedicine and Pharmacotherapy 132 (2020) 110850

[R5] Gajdosova, V., Lorencova, L., Kasak, P., Tkac, J., Electrochemical nanobiosensors for detection of breast cancer biomarkers, Sensors, 20 (2020),4022, pp. 1-37

[R6] Habli, Z., Alchamaa, W., Saab, R., Kadara, H., Khraiche, M.L., Circulating tumor cell detection technologies and clinical utility: Challenges and opportunities, Cancers 12 (2020),1930, pp. 1-30

[R7] Vajhadin, F., et al, Electrochemical cytosensors for detection of breast cancer cells, Biosensors and Bioelectronics 151 (2020)111984

[R8] Afreen, S., He, Z., Xiao, Y., Zhu, J.-J., Nanoscale metal-organic frameworks in detecting cancer biomarkers, Journal of Materials Chemistry B, 8(2020), pp. 1338-1349

[R9] Safarpour, H. et al., Optical and electrochemical-based nano-aptasensing approaches for the detection of circulating tumor cells (CTCs), Biosensors and Bioelectronics148 (2020)111833

2. In relation to the previous point, these recent review papers ([R1-R9]) focus on one or two sensing modules/techniques for CTC detection. On the other hand, the current manuscript attempts to review a wide range of the detection techniques, which can be the unique point. However, what is missing in the manuscript is a table summarizing each technique’s detection limit/sensitivity, range, and pros/cons. This will be highly useful for readers.

3. Introduction: The 2nd paragraph of the Introduction should be reorganized and rewritten. Rather than focusing on the details of the various sensing techniques, a more general view of the detection methodologies should be given in an attempt to answer the following questions. 1) Why/how are nanomaterials used in CTC detections? What physical/chemical/biological characteristics enhance the detection? You can also briefly list the most popular nanomaterials used for CTC detection here. 2) A variety of the sensing techniques can be mentioned with more generic terms. Here the authors should acknowledge the past reviews (see my first point) and state how this review paper advances the state of the knowledge.

4. I understand that the review paper cannot include all the relevant works reported in the literature. But some of the highly cited papers that are directly related to the topic were missing (see below). See the number of citations received for each paper according to Scopus. I wonder what criteria were used to select the papers reviewed in the manuscript (year published? citation numbers?). When writing a review paper, one should attempt to capture the works that were highly cited.  

[R10] Nguyen, A.H., Sim, S.J., Nanoplasmonic biosensor: Detection and amplification of dual bio-signatures of circulating tumor DNA, Biosensors and Bioelectronics 67 (2015) pp. 443-449 → 73 times

[R11] Shao, N., Wickstrom, E., Panchapakesan, B., Nanotube-antibody biosensor arrays for the detection of circulating breast cancer cells, Nanotechnology 19 (2008),465101 → 53 times

[R12] Li, X., et al., Simultaneous detection of MCF-7 and HepG2 cells in blood by ICP-MS with gold nanoparticles and quantum dots as elemental tags, Biosensors and Bioelectronics 90 (2017) pp. 343-348  → 48 times

[R13] Hong, W.Y., Jeon, S.H., Lee, E.S., Cho, Y., An integrated multifunctional platform based on biotin-doped conducting polymer nanowires for cell capture, release, and electrochemical sensing, Biomaterials 35 (2014), pp. 9573-9580 → 48 times

[R14] Zhang, X., Chen, B., He, M., Wang, H., Hu, B.,  Gold nanoparticles labeling with hybridization chain reaction amplification strategy for the sensitive detection of HepG2 cells by inductively coupled plasma mass spectrometry, Biosensors and Bioelectronics 86 (2016) pp. 736-740 → 43 times

5. Another thing a good review paper should present is a critical view of the existing techniques and outlook of the field. While some of these were briefly mentioned in Conclusions, I suggest that the authors expand a bit more and provide more specific examples of the future research directions and challenges associated with transitioning into the market. 

Author Response

Comment 1: Introduction: Is this the first review paper on the topic? Absolutely not. There have been many review papers on the detection of CTCs, which should be cited and critically reviewed in the manuscript. It is important to acknowledge the past review efforts and explicitly state what new perspectives the current paper brings to the community. From my quick search, I found the following review papers related to the nanomaterial-based CTC sensing just for Year 2020 and 2021. A lot more review papers can be found in the previous years. [R1] Li, X.-R., Zhou, Y.-G., Electrochemical detection of circulating tumor cells: A mini review, Electrochemistry Communications 124 (2021)106949. [R2] Farshchi, F., Hasanzadeh, M., Microfluidic biosensing of circulating tumor cells (CTCs): Recent progress and challenges in efficient diagnosis of cancer, Biomedicine and Pharmacotherapy, 134 (2021) 111153. [R3] Koo, K.M., Soda, N., Shiddiky, M.J.A., Magnetic nanomaterial–based electrochemical biosensors for the detection of diverse circulating cancer biomarkers, Current Opinion in Electrochemistry, 25 (2021) 100645. [R4] Fattahi, Z., Khosroushahi, A.Y., Hasanzadeh, M., Recent progress on developing of plasmon biosensing of tumor biomarkers: Efficient method towards early stage recognition of cancer, Biomedicine and Pharmacotherapy 132 (2020) 110850. [R5] Gajdosova, V., Lorencova, L., Kasak, P., Tkac, J., Electrochemical nanobiosensors for detection of breast cancer biomarkers, Sensors, 20 (2020),4022, pp. 1-37. [R6] Habli, Z., Alchamaa, W., Saab, R., Kadara, H., Khraiche, M.L., Circulating tumor cell detection technologies and clinical utility: Challenges and opportunities, Cancers 12 (2020),1930, pp. 1-30. [R7] Vajhadin, F., et al, Electrochemical cytosensors for detection of breast cancer cells, Biosensors and Bioelectronics 151 (2020)111984. [R8] Afreen, S., He, Z., Xiao, Y., Zhu, J.-J., Nanoscale metal-organic frameworks in detecting cancer biomarkers, Journal of Materials Chemistry B, 8(2020), pp. 1338-1349. [R9] Safarpour, H. et al., Optical and electrochemical-based nano-aptasensing approaches for the detection of circulating tumor cells (CTCs), Biosensors and Bioelectronics148 (2020)111833.

Response: We thank the reviewer for his/her comments. During the past decades, a lot of review articles have been reported on the enrichment and detection methodologies of CTCs [47-51]. For instance, Vajhadin’s group and Li’s group reviewed the progress in electrochemical detection of breast cancer cells and other CTCs [52,53]. Farshchi et al. reviewed recent progress and challenges in microfluidic biosensing of CTCs [54]. Koo et al. summarized the applications of magnetic nanomaterials in electrochemical detection of diverse circulating cancer biomarkers [55]. Besides, aptamer-based cytosensing approaches for the capture and detection of CTCs have been covered in several review papers [35,47,49,50]. However, to the best of our knowledge, few review papers focused on nanomaterials-based cytosensors for CTCs detection including both immunosensors and aptasensors [56,57]. In this review, we aim to give a comprehensive review about various nanoplatforms and highlight the different roles of nanomaterials in cytosensors. For discussion purposes, developments in CTCs detection are classified according to the sensing techniques, which covered mostly optical and electrochemical transducers. The optical cytosensors include fluorescence, colorimetry, surface-enhanced Raman scattering and chemiluminescence. Electrochemical methods are mainly represented by electrochemistry, electrochemiluminescence and photoelectrochemistry. We have revised the Introduction and cited these review papers.

Comment 2: In relation to the previous point, these recent review papers ([R1-R9]) focus on one or two sensing modules/techniques for CTC detection. On the other hand, the current manuscript attempts to review a wide range of the detection techniques, which can be the unique point. However, what is missing in the manuscript is a table summarizing each technique’s detection limit/sensitivity, range, and pros/cons. This will be highly useful for readers.

Response: This is a good suggestion. The analytical performances of the nanomaterials-based optical and electrochemical cytosensors for CTCs detection are compared in Table 1 and Table 2.

Comment 3: Introduction: The 2nd paragraph of the Introduction should be reorganized and rewritten. Rather than focusing on the details of the various sensing techniques, a more general view of the detection methodologies should be given in an attempt to answer the following questions. 1) Why/how are nanomaterials used in CTC detections? What physical/chemical/biological characteristics enhance the detection? You can also briefly list the most popular nanomaterials used for CTC detection here. 2) A variety of the sensing techniques can be mentioned with more generic terms. Here the authors should acknowledge the past reviews (see my first point) and state how this review paper advances the state of the knowledge.

Response: It is an excellent comment. We have written the introduction and added some sentences in red words to discuss the questions. Addressing the reviewers’ comments has improved the quality of our work.

Comment 4: I understand that the review paper cannot include all the relevant works reported in the literature. But some of the highly cited papers that are directly related to the topic were missing (see below). See the number of citations received for each paper according to Scopus. I wonder what criteria were used to select the papers reviewed in the manuscript (year published? citation numbers?). When writing a review paper, one should attempt to capture the works that were highly cited.  [R10] Nguyen, A.H., Sim, S.J., Nanoplasmonic biosensor: Detection and amplification of dual bio-signatures of circulating tumor DNA, Biosensors and Bioelectronics 67 (2015) pp. 443-449 → 73 times. [R11] Shao, N., Wickstrom, E., Panchapakesan, B., Nanotube-antibody biosensor arrays for the detection of circulating breast cancer cells, Nanotechnology 19 (2008),465101 → 53 times. [R12] Li, X., et al., Simultaneous detection of MCF-7 and HepG2 cells in blood by ICP-MS with gold nanoparticles and quantum dots as elemental tags, Biosensors and Bioelectronics 90 (2017) pp. 343-348 → 48 times. [R13] Hong, W.Y., Jeon, S.H., Lee, E.S., Cho, Y., An integrated multifunctional platform based on biotin-doped conducting polymer nanowires for cell capture, release, and electrochemical sensing, Biomaterials 35 (2014), pp. 9573-9580 → 48 times. [R14] Zhang, X., Chen, B., He, M., Wang, H., Hu, B.,  Gold nanoparticles labeling with hybridization chain reaction amplification strategy for the sensitive detection of HepG2 cells by inductively coupled plasma mass spectrometry, Biosensors and Bioelectronics 86 (2016) pp. 736-740 → 43 times.

Response: We thank the reviewer for his/her comments. We have cited the related papers in the revised manuscript.

Comment 5: Another thing a good review paper should present is a critical view of the existing techniques and outlook of the field. While some of these were briefly mentioned in Conclusions, I suggest that the authors expand a bit more and provide more specific examples of the future research directions and challenges associated with transitioning into the market. 

Response: We have revised the Introduction and Conclusion and improved the quality of this manuscript.

Reviewer 2 Report

This review tried to summarize the recent progress in cancer cell or cell marker detection using various signal amplification strategies based on different techniques. Although there are over 200 papers cited, some of the representative papers in each field still remains missing. Besides, the current version reads like an assembly of paper abstracts and conclusions, without any insightful comments or evaluation of each work, thus contributing quite little to the sensor field or each field (such as CL, FL, ECL, etc.). The reviewer would suggest the authors to focus on just one technique, such as ECL or FL, to deeply discuss the progress and problems existing in this field, with your own insights into this research area, which would be more attractive to readers.

Moreover, the language should be improved dramatically.

Author Response

Comment: This review tried to summarize the recent progress in cancer cell or cell marker detection using various signal amplification strategies based on different techniques. Although there are over 200 papers cited, some of the representative papers in each field still remains missing. Besides, the current version reads like an assembly of paper abstracts and conclusions, without any insightful comments or evaluation of each work, thus contributing quite little to the sensor field or each field (such as CL, FL, ECL, etc.). The reviewer would suggest the authors to focus on just one technique, such as ECL or FL, to deeply discuss the progress and problems existing in this field, with your own insights into this research area, which would be more attractive to readers. Moreover, the language should be improved dramatically.

Response: We thank the reviewer for his/her comments. We have revised the paper and cited more references. We believe that the quality of this manuscript has been greatly improved after addressing the reviewers’ comments.

Reviewer 3 Report

this is a review that looks at the utility of nanomaterials to detect the presence of circulating tumour cells, an important group of cells that serves as the precursor to rapidly growing tumours. this is a timely review and is an important contribution to the area. the manuscript is succinctly written and covers the recent progress in this rapidly growing field adequately. suitable diagrams and images are used to support their description, which is very helpful. the only thing that i suggest to improve this already very good manuscript is to touch very briefly on how there are other methods to detect CTCs before zeroing in on nanomaterials. for instance, microfluidics seems to be an important area given the recent literature in this area:

"Sorting Technology for Circulating Tumor Cells Based on Microfluidics" ACS Comb. Sci. 2020, 22, 12, 701–711.

“Recent advances in microfluidic methods in cancer liquid biopsy” Biomicrofluidics 2019, 13, 041503.

"Fast and efficient microfluidic cell filter for isolation of circulating tumor cells from unprocessed whole blood of colorectal cancer patients"  Scientific Reports 2019, 9, 8032.

Author Response

Comments: this is a review that looks at the utility of nanomaterials to detect the presence of circulating tumour cells, an important group of cells that serves as the precursor to rapidly growing tumours. this is a timely review and is an important contribution to the area. the manuscript is succinctly written and covers the recent progress in this rapidly growing field adequately. suitable diagrams and images are used to support their description, which is very helpful. the only thing that i suggest to improve this already very good manuscript is to touch very briefly on how there are other methods to detect CTCs before zeroing in on nanomaterials. for instance, microfluidics seems to be an important area given the recent literature in this area: "Sorting Technology for Circulating Tumor Cells Based on Microfluidics" ACS Comb. Sci. 2020, 22, 12, 701–711. “Recent advances in microfluidic methods in cancer liquid biopsy” Biomicrofluidics 2019, 13, 041503. "Fast and efficient microfluidic cell filter for isolation of circulating tumor cells from unprocessed whole blood of colorectal cancer patients"  Scientific Reports 2019, 9, 8032.

Response: We thank the reviewer for his/her comments. The MBs-based and microfluidic techniques can be easily integrated with various optical and electrochemical analysis methods, promoting the development of novel biosensors. We have rewritten the Introduction to discuss the application of microfluidics in the separation and detection of CTCs. In this review, we aim to give a comprehensive review about various nanoplatforms and highlight the different roles of nanomaterials in cytosensors. For discussion purposes, developments in CTCs detection are classified according to the sensing techniques, which covered mostly optical and electrochemical transducers. The optical cytosensors include fluorescence, colorimetry, surface-enhanced Raman scattering and chemiluminescence. Electrochemical methods are mainly represented by electrochemistry, electrochemiluminescence and photoelectrochemistry. Other cytosensors for the detection of CTCs have been discussed in Part 2.8.

Round 2

Reviewer 1 Report

The authors have sufficiently addressed the reviewer's comments and incorporated them into the revised manuscript. The manuscript is much closer to the publication. I recommend it for publication after careful proofreading.

Author Response

We thank the reviewer for his/her positive comments. We have checked carefully and revised the manuscript again.

Reviewer 2 Report

Generally, the paper was improved to a certain extent, however, some major parts still needs improvement, such as the comments or insights into each work (this part should be well addressed before publication). The current version still reads like the assembly of paper abstracts. The other problem is that it is hard to distinguish this review to other reviews, although the author mentioned some points in the introduction. The specialty of this work should be well addressed.

Besides, some minor points should be revised:

1, some more representative papers in this field should be cited and well discussed, such as but not limited to:

Analytical Chemistry  201890(20), 12284-12291; Analytical Chemistry  202092(10), 7354-7362; Analytical Chemistry, 2018, 90 (18), 10858–10864. ; Analytical Chemistry,  202193(22), 7925-7932.; Analytical Chemistry  202092(22), 15120-15128. 

Some more recent works in this field published in 2021, should be cited as well.

2, the language should be polished by some language agency.

Author Response

Comment 1: Generally, the paper was improved to a certain extent, however, some major parts still needs improvement, such as the comments or insights into each work (this part should be well addressed before publication). The current version still reads like the assembly of paper abstracts. The other problem is that it is hard to distinguish this review to other reviews, although the author mentioned some points in the introduction. The specialty of this work should be well addressed.

Response: We thank the reviewer for his/her comments. We have added the comments on each work and revised the manuscript carefully. The added sentences have been shown in red words. Recently, a few review articles have been reported on the enrichment and detection methodologies of CTCs. For instance, Khademhossieni’s group and Zhou’s group reviewed the progress in electrochemical detection of breast cancer cells and other CTCs. Farshchi et al. reviewed recent progress and challenges in microfluidic biosensing of CTCs. Koo et al. summarized the applications of magnetic nanomaterials in electrochemical detection of diverse circulating cancer biomarkers. Besides, aptamer-based cytosensing approaches for the capture and detection of CTCs have been covered in several review papers. However, to the best of our knowledge, there are few reports focused on nanomaterials-based cytosensors for CTCs detection including both immunosensors and aptasensors. In this review, we aim to give a comprehensive review about various nanoplatforms and highlight the different roles of nanomaterials in cytosensors. For discussion purposes, developments in CTCs detection are classified according to the sensing techniques, which covered mainly optical and electrochemical transducers.

Comment 2: Besides, some minor points should be revised: 1, some more representative papers in this field should be cited and well discussed, such as but not limited to: Analytical Chemistry  2018, 90(20), 12284-12291; Analytical Chemistry  2020, 92(10), 7354-7362; Analytical Chemistry, 2018, 90 (18), 10858–10864.; Analytical Chemistry,  2021, 93(22), 7925-7932.; Analytical Chemistry  2020, 92(22), 15120-15128. Some more recent works in this field published in 2021, should be cited as well.

Response: We have discussed the works and cited the papers published in 2021.

Comment 3: the language should be polished by some language agency.

Response: We have revised the manuscript carefully and the language has been polished by an English speaker.

Addressing the reviewer’s comments has improved the quality of our work. Thank you very much again for your consideration of publication of our manuscript.